# Rescuing the bacterial replisome at a nick requires recombinational repair and helicase reloading

Charles Winterhalter ✉, Kathryn J. Stratton, Stepan Fenyk & Heath Murray ✉

DNA damage occurs in all cells and must be repaired to maintain genome integrity. Many DNA lesions are targeted for removal by repair systems that excise the damage, thereby generating a temporary single-strand discontinuity in the chromosome. If DNA repair has not been completed prior to a round of genome duplication, the single-strand discontinuity (nick or gap) can be converted to a double-strand break (DSB) by an oncoming replication fork. Because the genomic location of nucleobase damage is stochastic, investigating the fate of replication machinery (replisome) at DNA repair sites with single-strand discontinuities has been limited. Here we address this issue by expressing Cas9 nickases in *Bacillus subtilis* to create site-specific single-strand discontinuities in a bacterial chromosome. We find that a nick in either leading or lagging strand arrests DNA replication, while the fate of the replicative helicase is distinct and depends upon the strand nicked. Genetic, biochemical, and single cell analyses indicate that replisome/nick encounters generate a single-end DSB which requires recombinational repair to enable PriA-dependent replication restart. Together this work defines the physiologically relevant pathway used by *B. subtilis* to reinitiate DNA synthesis following replication fork inactivation at a single-strand discontinuity.

Intracellular DNA can become damaged and/or distorted through a multitude of endogenous and exogenous sources (e.g., post-replication mismatches, base oxidation, cyclobutane pyrimidine dimers)[1,2]. Damaged and distorted DNA can be mutagenic, resulting in permanent changes to genetic information of progeny. To avoid these negative consequences in genome fidelity, specialised DNA repair systems recognise and remove the damaged (base/nucleotide excision repair) or distorted (mismatch repair) DNA.

As part of their repair mechanism, these systems excise the damaged or mismatched base, causing the formation of a nick or single-strand gap in the affected DNA strand[3]. Critically, if a replication fork progresses towards such a single-strand discontinuity before repair is complete, the encounter can generate a single-end double-strand break (seDSB)[4]. Elegant experiments using bacteriophage λ

demonstrated that an engineered nick in either strand of the viral genome generated replication-dependent DSBs (Fig. 1A)[5]. While active DNA replication was necessary to generate DSBs at a nick[5], the fate of the bacterial DNA replication machinery (replisome) at single-strand discontinuities has not been determined.

The replisome is a multicomponent complex of proteins and enzymes that unwinds the chromosome and coordinates synthesis of new DNA polymers on the leading and lagging strands[6,7]. The most stable component of the replisome, and most pivotal for its processivity, is helicase[8,9]. Cellular replicative helicases are toroid enzymes that unwind DNA by encircling and translocating along a single client nucleic acid strand, acting as a wedge to disrupt base pairing[10]. Studies of bacterial replisomes in vitro and in vivo have revealed that the machinery is intrinsically dynamic, with most factors observed to bind

Centre for Bacterial Cell Biology, Biosciences Institute, Newcastle University, Newcastle Upon Tyne, UK. ✉e-mail: Charles.winterhalter@newcastle.ac.uk; Heath.murray@newcastle.ac.uk

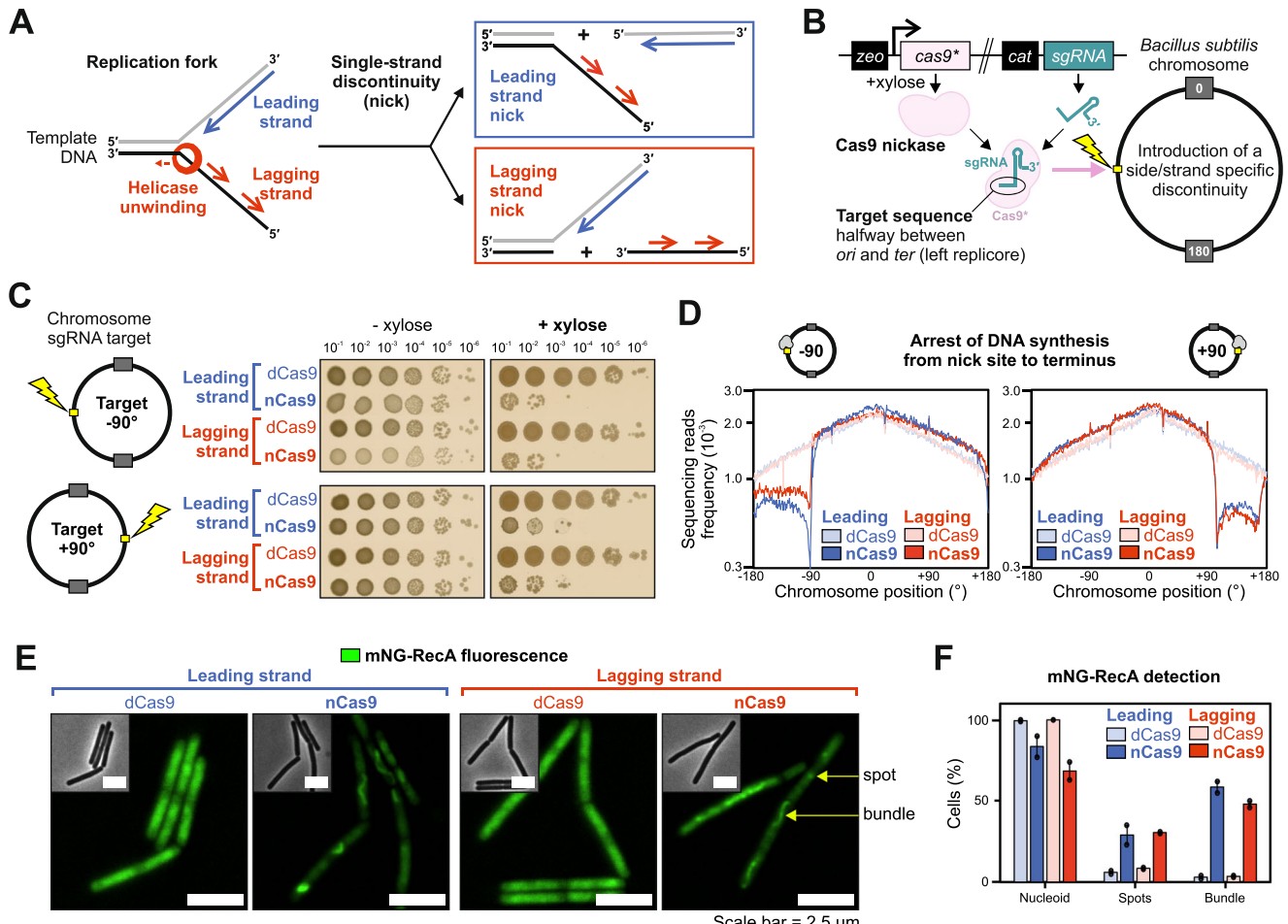

**Fig. 1 | Single-strand discontinuities arrest DNA synthesis and generate DNA damage. A** Diagram illustrating the impact of leading and lagging strand discontinuities on a replication fork, with the bacterial replicative helicase translocating along the lagging strand template. **B** Genetic system employed to create a site-specific single-strand discontinuity in the *B. subtilis* chromosome. Black boxes correspond to antibiotic resistance markers, pink to Cas9 variants and green to sgRNA. Chromosome annotations refer to the replication origin (0°), terminus (180°) and sgRNA target sequence (yellow box). **C** Spot-titre analysis of strains engineered to express (+ xylose) either nCas9 or dCas9. Individual sgRNAs were designed to target the leading or lagging strand template at loci located at either -90° (top panel) or +90° (bottom panel). **D** MFA using whole genome sequencing of strains engineered to express (+ xylose) either nCas9 or dCas9. Individual sgRNAs were designed to target the leading or lagging strand template at loci located at

either -90° (left panel) or +90° (right panel). The frequency of sequencing reads was plotted against genome position. **E** Fluorescence microscopy images of the DNA damage reporter mNG-RecA in strains engineered to express (+ xylose) either nCas9 or dCas9 (sgRNAs target -90°). Fluorescent RecA organises into bundles in the presence of DNA damage. Green signal corresponds to mNG-RecA fluorescence, grey scale images show corresponding phase contrast images. Scale bar indicates 2.5 μm. **F** Quantification of fluorescence microscopy images of the DNA damage reporter mNG-RecA in strains engineered to express (+ xylose) either nCas9 or dCas9 (sgRNA target -90°). Detection of spots and bundles corresponds to distinct foci or filament-like structures annotated in (**E**). Error bars indicate standard deviation and circles overlaid on bars correspond to two biological replicates. Source data are provided as a Source Data file.

and unbind interaction partners multiple times per replication cycle[11,12]. This behaviour enables the replisome to overcome a range of potential roadblocks and remain associated with the template, competent to continue DNA synthesis[13]. However, recent studies in eukaryotic systems have indicated that single-strand discontinuities generate DSBs and arrest DNA replication by impairing helicase progression[14,15].

Most bacterial chromosomes contain a single replication origin (*oriC*) that promotes assembly of two independent replication forks for bidirectional DNA synthesis[16,17]. Consequently, replisomes initiated from *oriC* must continue until they complete replication of their respective chromosome arm[18–21]. In contrast, eukaryotic chromosomes contain multiple replication origins, therefore no single replication fork is essential to complete genome duplication[15,16,22]. If a bacterial replisome encounters a single-strand discontinuity and generates a DSB, it has been proposed that rescue of the broken chromosome occurs through homologous recombination and that reassembly of

the replisome follows helicase loading at the repaired replication fork[23].

Determining how bacteria react and respond when a replication fork encounters a single-strand discontinuity is crucial to understanding how these organisms ensure genome stability. Here we have investigated this question by expressing Cas9 nickases (nCas9) in the model bacterium *Bacillus subtilis*. The results show that a nick located in either template strand blocks downstream DNA synthesis. Resumption of DNA replication is dependent upon a core set of recombinational repair proteins and the PriA-dependent replication restart system to reload the replicative helicase.

## Results

### Single-strand discontinuities in a bacterial chromosome arrest DNA synthesis

Damaged and mismatched nucleobases can be located randomly throughout a genome, making it a challenge to directly analyse

conflicts between DNA replication and repair. To mimic the formation of a single-strand discontinuity produced during DNA repair, an experimental system capable of generating a nick was developed in *B. subtilis* using Cas9 nuclease variants and single-guide RNAs (sgRNA)[14,15,22,24]. Cas9 carrying single amino acid substitutions (Cas9[D10A] or Cas9[H840A]) creates nickases that can cleave the phosphodiester bond of one DNA strand (nCas9), while the double mutant (Cas9[D10A/H840A]) creates a catalytically inactive enzyme (dCas9)[25]. Each *cas9* allele was integrated into the *B. subtilis* chromosome under the control of a xylose-inducible promoter (Fig. 1B)[26]. The location of sgRNA hybridisation is indicated by coordinates between the origin (0°) and terminus (~180°) on either the right (+) or left (-) arm of the chromosome. To avoid complications arising from potential activation of prophage following DNA damage in the laboratory strain of *B. subtilis*, a derivative lacking these genetic elements (Δ6) was used throughout this study[27].

Cell viability following nicking was quantified using spot-titre assays, where strains are grown to saturation in liquid medium without xylose, before samples were serially diluted and plated onto solid medium with or without xylose to enumerate the number of colony forming units (CFUs). It was observed that the Cas9[D10A] nickase elicits a stronger phenotype than Cas9[H840A] upon induction with xylose (Supplementary Fig. 1A, B)[25]. To achieve comparable nicking efficiency at different sites in the chromosome, the *cas9[D10A]* allele was used throughout the remainder of this study, in conjunction with distinct sgRNAs targeted to non-essential regions of the genome. Using sgRNAs to target either leading or lagging strands at loci on the left or right chromosome arms, it was observed that expression of nCas9 resulted in severe growth inhibition (Fig. 1C, Supplementary Fig. 1C). In contrast, expression of dCas9 targeted to the same locations did not affect the number of CFUs (Fig. 1C, Supplementary Fig. 1D), indicating that DNA nicking is necessary to cause growth inhibition.

Next chromosome replication was investigated using whole genome marker frequency analysis (MFA). Expression of nCas9 was induced for one hour in exponentially growing cultures. Genomic DNA (gDNA) was harvested for next-generation sequencing, reads were mapped onto the reference genome sequence, and data is displayed with the chromosome origin (*oriC*) at the centre of the graphs (Supplementary Fig. 2). Introduction of nicks in either the leading or the lagging strand resulted in a drop in sequencing coverage from the nick site to the terminus region, whereas expression of dCas9 with the same sgRNAs did not appear to inhibit DNA replication (Fig. 1D). Copy number at the nick position was low in the presence of nCas9, potentially reflecting DNA degradation[28]. Consistent with a DNA replication-dependent phenotype, spot-titre analyses showed that nick induction in late-exponential phase does not result in loss of viability, whereas induction in early- exponential phase results in severe growth defects (Supplementary Fig. 1E). Therefore, because cells expressing nCas9 are unable to synthetise DNA downstream of the nick, incomplete chromosome synthesis can explain the strong growth inhibition observed in spot-titre assays (Fig. 1C).

Previous results indicated that a bacterial replication fork encountering a single-strand discontinuity generates a DSB[5]. To determine whether DNA damage was generated following nCas9 induction, localisation of an ectopic mNeonGreen-RecA reporter (mNG-RecA) was observed in live cells using fluorescence microscopy (endogenous *recA* is present in these merodiploid strains). It has been shown in *B. subtilis* that following DNA damage fluorescently labelled RecA recombinase assembles into bundles[29–31]. Consistent with the literature, in unstressed cells mNG-RecA is observed evenly distributed over the nucleoid (Supplementary Fig. 3A)[32]. Following expression of nCas9 for 90 minutes during exponential growth, mNG-RecA bundles were detected in 59% of cells when the leading strand template was nicked and 48% of cells when the lagging strand template was nicked (Fig. 1E, F, Supplementary Fig. 3B). In contrast, expression of dCas9 did

not significantly promote mNG-RecA bundling (Fig. 1E, F, Supplementary Fig. 3B), validating that mNG-RecA is a functional reporter for DNA damage. These results are consistent with the model that the bacterial replisome generates DNA damage upon encountering a nick in either strand of the parental DNA duplex.

## Single-strand discontinuities in a bacterial chromosome inactivate the replisome

MFA indicated that following expression of nCas9, DNA synthesis does not proceed beyond the nick. We wondered whether the replisome is arrested at a single-strand discontinuity or whether it is inactivated. If the latter event occurs, then the bacterial replication restart system would be required to reload helicase and promote replisome assembly[33,34]. In *B. subtilis* there is only one known replication restart system composed of the essential proteins PriA and the helicase loaders DnaD, DnaB, and DnaI (note that in *B. subtilis* the replicative helicase is DnaC)[35–39].

Chromatin immunoprecipitation (ChIP) followed by quantitative PCR (qPCR) was used to determine the enrichment of helicase and replication restart proteins surrounding a nick in either the leading or lagging strand template. Following induction of nCas9 for one hour in exponentially growing cells, samples were harvested for ChIP. Relative protein enrichment proximal to the nick was normalised to a control locus located on the opposite chromosome arm equidistant from *oriC* (+90°, Fig. 2A). ChIP showed that helicase is specifically enriched upstream of nicks located in either template strand (Fig. 2B, Supplementary Fig. 4). This upstream helicase enrichment was not detected when dCas9 was expressed, indicating that the Cas9-sgRNA nucleoprotein complex alone does not perturb replication fork progression (Fig. 2B, Supplementary Fig. 4).

Interestingly, significant enrichment of helicase was observed downstream of the nick site when cleavage occurred in the template used for leading strand synthesis (Fig. 2B, Supplementary Fig. 4). The bacterial replicative helicase translocates in the 5′→3′ direction along the template used for lagging strand synthesis[16]. Therefore, ChIP results suggest that when the lagging strand template is nicked helicase runs-off the DNA substrate, whereas when the leading strand template is nicked helicase remains associated with the template DNA and progresses past the single-strand discontinuity (Fig. 2B). Consistent with the observed downstream helicase enrichment (Fig. 2A, B) and replisome inactivation (Fig. 1D), reconstituted helicase assays in vitro have shown that when the bacterial enzyme encounters a leading strand nick it transitions to encircling double-stranded DNA (dsDNA) and slides along the double-helix without further unwinding activity[40].

The enrichment of helicase upstream of nick sites is consistent with either helicase stalling or helicase reloading via the replication restart pathway. To further investigate these models, ChIP was used to probe the presence of replication restart proteins surrounding the nicks. Following nCas9 induction, enrichment of PriA, DnaD, DnaB and DnaI was detected upstream (but not downstream) of the sgRNA target sequences, while these proteins were absent from this region when dCas9 was expressed (Fig. 2C-F). These results suggest that nicks in either the leading or lagging strand template elicit the replication restart pathway to reload the replicative helicase, and they are consistent with models where helicase either runs off the template (lagging strand template nick) or translocates downstream (leading strand template nick) in an inactive state (Fig. 2G).

## A weakened nCas9 system permits DNA replication restart

While replication restart proteins were enriched upstream of nick sites (Fig. 2B–F), viability assays and MFA both indicated that downstream DNA synthesis was inhibited (Fig. 1C, D). We speculated that persistent nCas9 cleavage activity competes with productive DNA replication restart. Therefore, to allow interrogation of processes occurring

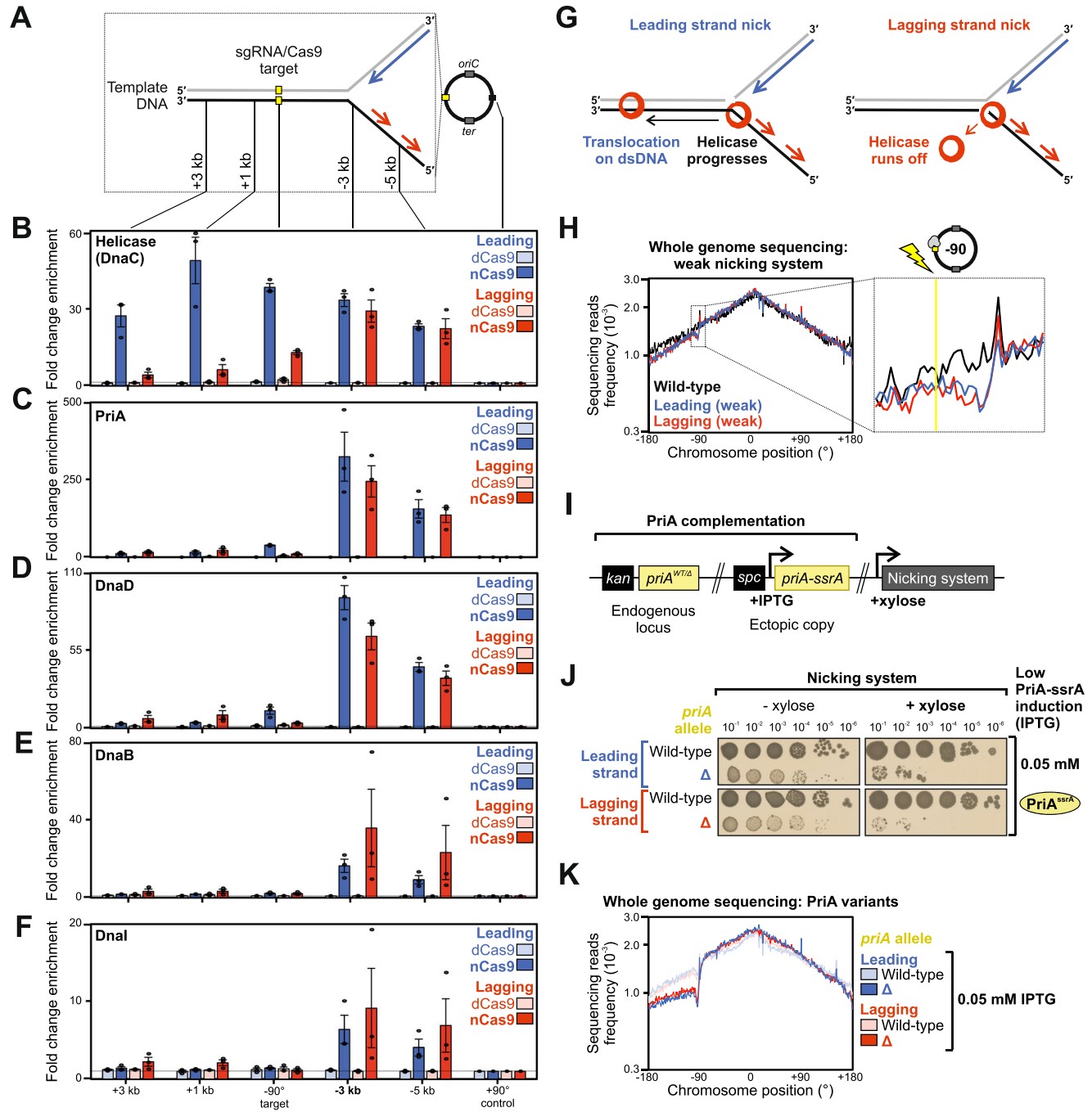

**Fig. 2 | Replication restart is necessary to resume DNA synthesis following replisome inactivation at a single-strand discontinuity. A** Diagram illustrating genomic loci probed via ChIP-qPCR at either -90° or +90°. The sgRNA targeted Cas9 protein variants to the locus at -90°. Lagging strand and leading strand templates are represented in black and grey lines, respectively (numbers refer to strand polarity). The blue arrow indicates leading strand synthesis and red arrows indicate lagging strand synthesis. **B–F** ChIP-qPCR analyses of helicase (DnaC) and replication restart proteins (PriA, DnaD, DnaB, DnaI) in strains engineered to express either nCas9 or dCas9. The sgRNAs targeted Cas9 proteins to a locus located at -90° and protein enrichment was normalised to the control locus at +90°. Error bars indicate the standard error of the mean and circles overlaid on bars

correspond to three biological replicates. **G** Diagram illustrating the fate of helicase (red circle) upon encountering a single-strand discontinuity in either the leading or lagging strand template. **H** MFA using whole genome sequencing in strains engineered to express low levels of sgRNAs and Cas9 proteins (weak nicking system). The frequency of sequencing reads was plotted against genome position. **I** Genetic system employed to express limiting levels of PriA in strains harbouring the weak nicking system. **J** Spot-titre analysis of strains engineered to express nCas9 (+ xylose) with either high or limiting PriA. sgRNAs were targeted to a locus at -90°. **K** MFA using whole genome sequencing in strains engineered to express (+ xylose) nCas9 with either high or limiting PriA. The frequency of sequencing reads was plotted against genome position. Source data are provided as a Source Data file.

downstream of DNA cleavage and replisome inactivation, a weaker nCas9 system was developed. An *ssrA* degradation tag[41] was directly fused to *cas9*[D10A] to promote nCas9-ssrA proteolysis, and expression of both *nCas9-ssrA* and *sgRNAs* were placed under the control of xylose-inducible promoters (Supplementary Fig. 5A). In this background,

induction of nCas9 did not significantly inhibit cell viability or alter chromosome content (Supplementary Fig. 5B, C), and MFA showed that DNA synthesis continued after a one-hour xylose induction, consistent with low tolerable levels of nickase activity (Fig. 2H, note a reduction in read depth upstream of nickase target sites, likely

reflecting recombinational repair). This dampened nicking system was employed to determine genes essential for replication restart.

## PriA is required to restart DNA replication at single-strand DNA discontinuities

The master replication restart factor PriA is essential for growth in *B. subtilis*. To ascertain whether PriA-dependent replication restart is required following chromosome nicking, an IPTG-dependent titratable PriA complementation system was developed (Supplementary Fig. 6A). Here conditions were established to express PriA-ssrA at a low level, sufficient to maintain viability while limiting the amount of PriA activity within a cell (Supplementary Fig. 6B, C). When nCas9-ssrA and sgRNA were induced under conditions with limiting PriA-ssrA, spot-titre analyses showed that cell growth is inhibited (Fig. 2I, J). As expected, higher expression of PriA-ssrA rescued viability (Supplementary Fig. 6D). Following induction of nCas9 and sgRNA for one hour during exponential growth, MFA showed that limiting levels of PriA-ssrA could not sustain chromosome synthesis downstream of the nick site (Fig. 2K). These data indicate that PriA is essential to restart DNA replication at a single-strand DNA discontinuity, consistent with the observed enrichment of helicase loading proteins (Fig. 2D–F) and with the model that replisomes encountering a nick are inactivated.

## A core set of recombinational repair proteins are necessary for DNA replication past a single-strand discontinuity

Genetic and cytological data suggest that when a replisome encounters a nick, it generates a substrate with ssDNA that is sensed by RecA (Fig. 1C–E). Evidence from several experimental systems indicate that under these conditions a single-end DSB is formed (seDSB)[28,42]. In bacteria, it has been proposed that a replication-dependent DSB is repaired through homologous recombination with the intact sister chromosome, thereby repairing the lesion and recreating a replication fork to resume DNA synthesis[43–46]. To determine the factors required for chromosome replication following nCas9 nicking, a targeted reverse genetic analysis was employed, focused on genes previously implicated in homologous recombination and other DNA repair processes (Supplementary Data 1). Using this approach, spot-titre analyses revealed that eight non-essential genes were required for normal growth upon nick induction: DNA end-processing genes *addA/addB* and recombinational repair genes *recF/recO/recR/recA/recG/recU* (Fig. 3A, Supplementary Fig. 7A). Functional complementation of each synthetic lethal deletion confirmed that these gene products are necessary to support growth in cells expressing nCas9 (Supplementary Fig. 7B). Interestingly, the screen revealed that several factors implicated in DNA repair are not essential for viability under these conditions, such as those classed for end-processing (RecQ, RecS, RecJ, Supplementary Fig. 8A)[28,47], maintaining chromosome structure ("SMC-like" RecN, SbcCD, SbcEF, Supplementary Fig. 8B)[48,49], and recombination mediation (RecD2, RecX, Supplementary Fig. 8C)[50–52]. Gene products involved in Holliday junction migration (RuvA, RuvB)[47] displayed a small colony phenotype but not a loss of CFUs (Supplementary Fig. 8D), suggesting they play a notable yet less critical role in the repair pathway.

Focusing on the eight mutants that produced the most severe growth defects, MFA was used to analyse DNA replication following induction of nCas9 and sgRNAs in exponentially growing cells, note several of the deletion mutants display altered replication profiles near the origin or terminus, consistent with previous studies (Supplementary Fig. 9)[53–55]. The results showed that DNA end-processing (*addA/addB*) and recombination mutants (*recF/recO/recR/recA/recG*) are defective in chromosome replication downstream of the nick site, with the only exception being *recU* (Fig. 3B and Supplementary Figs. 10, 11, note MFA profiles reflect distinct mutant backgrounds). Together, these observations support the recombinational repair model for rescuing a collapsed replication fork[56]. Moreover, the

requirement for the AddAB helicase/nuclease complex, which engages linear DNA substrates at a nearly blunt end[57], strongly suggests that bacterial replisomes encountering a nick generate a seDSB[58,59].

Based on the proposed activities of these proteins, the following model was developed (Fig. 3C). AddAB helicase/nuclease processes a broken replication fork at a seDSB to generate a single-stranded substrate with a 3′-end required for recombination (Fig. 3C(I–III))[23,58,59]. The single-stranded DNA (ssDNA) is recognised and bound by the single-stranded binding protein (SSB), which must be removed by the RecFOR system to allow RecA filament formation (Fig. 3C(IV–V))[23,60–62]. RecA is the master bacterial recombinase that binds to ssDNA with a free 3′-end, identifies homologous DNA on the intact sister chromosome, and performs strand invasion and exchange (Fig. 3C(V–VI))[63]. RecG is a helicase that remodels branched DNA substrates, likely acting at the Holliday junction or at the replication fork (these are not mutually exclusive) (Fig. 3C(VII))[64,65]. RecU is a structure specific nuclease involved in cleaving the Holliday junction generated by RecA (Fig. 3C(VII))[66].

## AddAB helicase activity is necessary for recombinational repair at a nick

The genetic analysis above indicated that the AddAB helicase/nuclease was responsible of processing the seDSB (Fig. 3). While both AddA and AddB subunits harbour nuclease activity, the two enzymes cleave distinct strands of the DNA duplex following unwinding by the AddA helicase (Fig. 4A)[59]. Whether all AddAB activities are required for recombinational repair in live cells remained unclear.

To determine whether AddAB was acting directly on broken replication forks generated by replisome inactivation at a nick, enrichment of AddAB was investigated using ChIP. AddA and AddB were independently fused to the fluorescent protein mNeonGreen (mNG), which here was used as an epitope tag. The results showed that mNG-AddA and mNG-AddB are maximally enriched 2 kb upstream of nicks, consistent with a direct role in end-processing under these conditions (Fig. 4B). The enzymatic activities of AddA and AddB were then reduced through substitution of catalytic residues (Fig. 4A)[59]. Spot-titre analyses showed that the helicase defective AddA$^{K36A}$ (AddA$^{ΔHEL}$) is unable to sustain bacterial growth upon induction of nicks, whereas nuclease defective variants of either AddA$^{D1172A}$ (AddA$^{ΔNUC}$) or AddB$^{D961A}$ (AddB$^{ΔNUC}$) grew comparably to a strain encoding wild-type AddAB (Fig. 4C). MFA confirmed that DNA synthesis is blocked downstream of a nick in the strain expressing AddA$^{ΔHEL}$, whereas the strains expressing AddA$^{ΔNUC}$, AddB$^{ΔNUC}$ or AddA$^{ΔNUC}$AddB$^{ΔNUC}$ were comparable to wild-type (Fig. 4D, Supplementary Fig. 12A, B). An immunoblot showed that the AddA$^{ΔHEL}$ variant is stably expressed (Supplementary Fig. 12C), indicating that AddAB helicase activity alone is necessary for the enzyme to process the seDSB.

Since AddAB nuclease activity was dispensable for recombinational repair, we wondered whether another nuclease might fulfil this role. RecJ is a single-strand DNA-specific exonuclease that has been observed to co-localise with the replisome in *B. subtilis*[67]. Deletion of *recJ* in conjunction with either *addA$^{ΔNUC}$*, *addB$^{ΔNUC}$*, or *addA$^{ΔNUC}$addB$^{ΔNUC}$* did not reduce the number of CFU following nicking (Supplementary Fig. 12D), indicating that RecJ nuclease activity is not required. Based on these observations, it appears that under these conditions AddAB helicase activity and an unspecified nuclease promote end-processing of the seDSB to generate a 3′-tail.

## SSB recruitment requires AddAB helicase activity

The requirement of the RecFOR system for recombinational repair (Fig. 2) implied that SSB is bound to ssDNA at the seDSB. To interrogate this hypothesis, recruitment of SSB was probed using ChIP following induction of nCas9 and sgRNAs in exponentially growing cells. The results showed that SSB is enriched upstream of a leading or a lagging strand nick (Fig. 5A), and that this enrichment was dependent upon

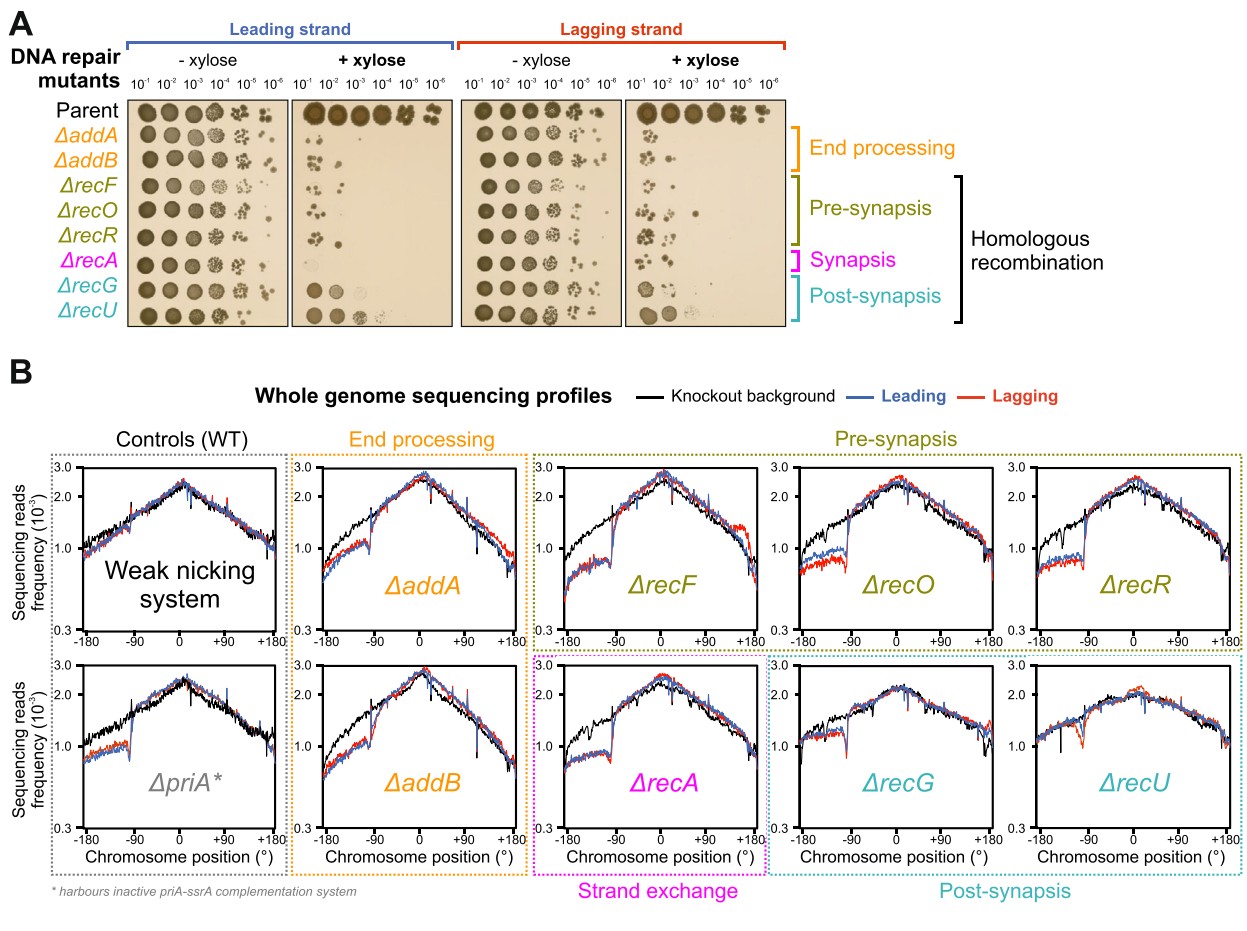

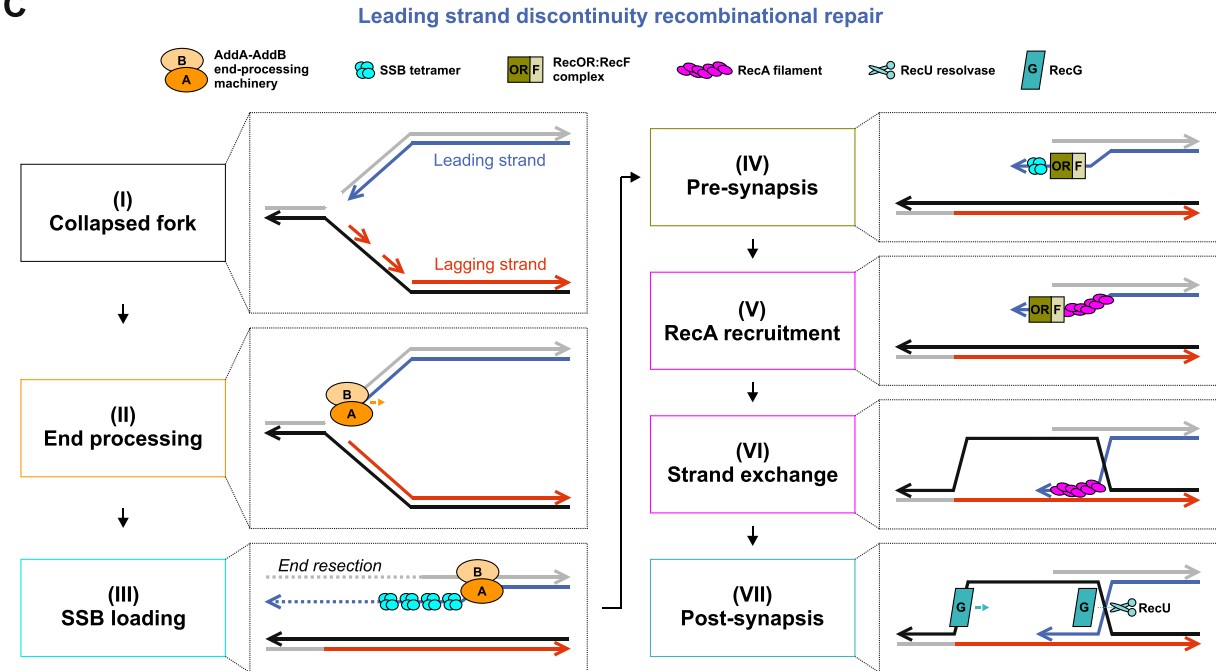

**Fig. 3 | The repair of single-strand discontinuities requires recombination.**
**A** Spot-titre analysis of recombinational repair mutants in strains engineered to express (+ xylose) nCas9. sgRNAs were targeted to a locus at -90°. The parental nicking strain control is shown above. **B** MFA using whole genome sequencing of recombinational repair mutants in the presence or absence of the weak nCas9 system. The frequency of sequencing reads was plotted against genome position. **C** Diagram illustrating proposed molecular events required for recombinational repair of a seDSB created after a replication fork encounters a single-strand discontinuity in the leading strand template. Source data are provided as a Source Data file.

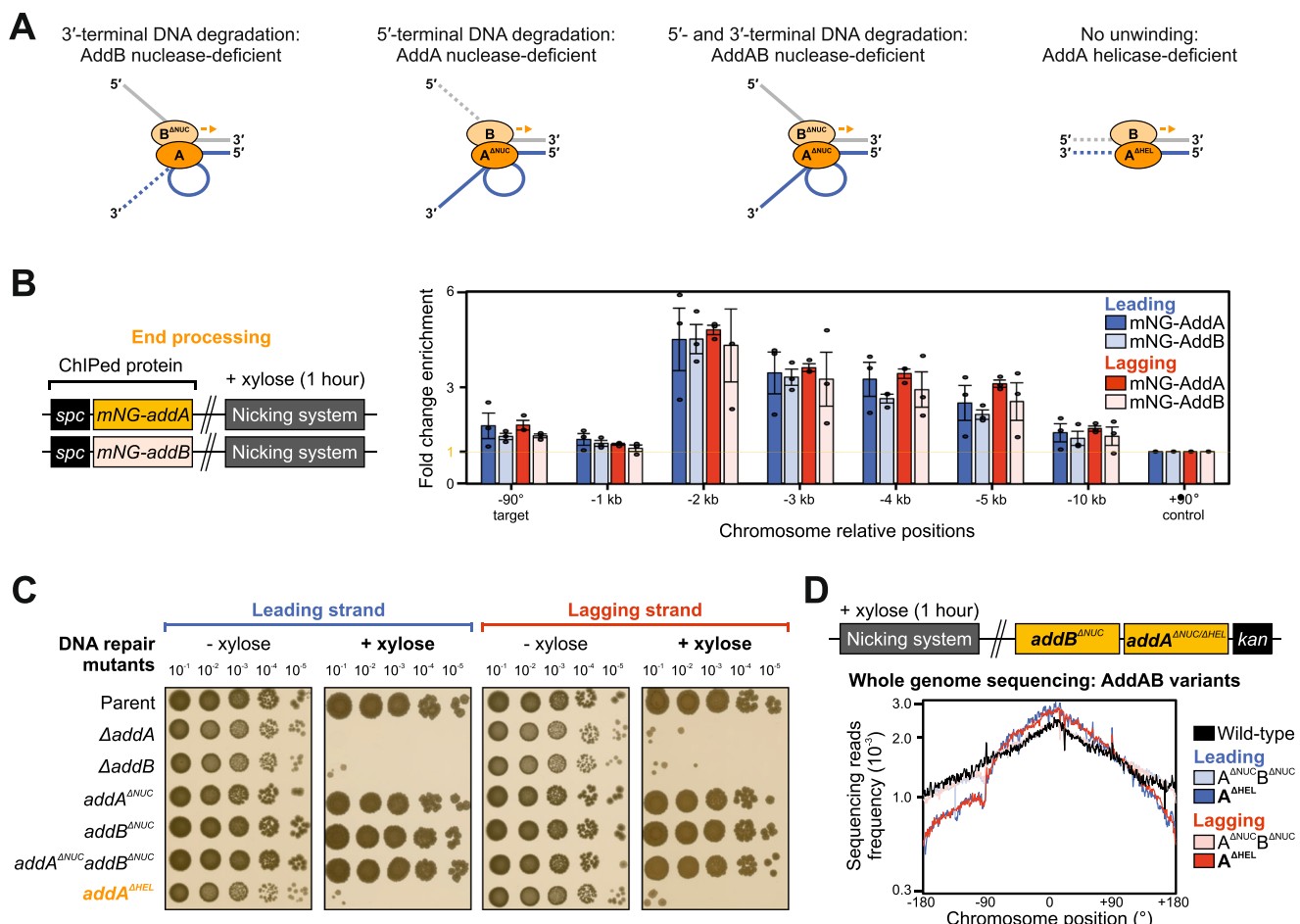

**Fig. 4 | The helicase activity of AddAB is necessary and sufficient to support DNA replication past a nick. A** Diagram illustrating differential resection of linear DNA in the presence of either the AddA or AddB helicase/nuclease activities. Dotted lines indicate nuclease cleavage. **B** ChIP-qPCR analyses of mNG-AddA and mNG-AddB in strains engineered to express nCas9. Left panel shows the genetic system employed in this experiment. The sgRNAs targeted nCas9 to a locus located at -90° and protein enrichment was normalised to the control locus at +90°. Error bars indicate the standard error of the mean and circles overlaid on bars correspond to three biological replicates. **C** Spot-titre analysis in strains with AddAB variants (AddA$^{\Delta HEL}$, AddA$^{\Delta NUC}$, AddB$^{\Delta NUC}$, AddA$^{\Delta NUC}$AddB$^{\Delta NUC}$) engineered to express (+ xylose) nCas9. sgRNAs were targeted to a locus at -90°. The parental nicking strain control is shown above. **D** MFA using whole genome sequencing of strains with either AddA$^{\Delta HEL}$ or AddA$^{\Delta NUC}$AddB$^{\Delta NUC}$ engineered to express nCas9. Top panel shows the genetic system employed in this experiment. The frequency of sequencing reads was plotted against genome position. Source data are provided as a Source Data file.

AddAB (Fig. 5B). Further analysis of AddAB showed that only helicase activity was required to promote SSB enrichment upstream of a nick (Fig. 5C). Together these results support the role of AddAB to unwind a seDSB and provide a ssDNA substrate for SSB.

AddAB helicase/nuclease activity is thought to be modulated by interaction of the enzyme with ssDNA binding motifs termed *chi* sites scattered throughout the genome[59,68,69]. The locations of *B. subtilis chi* sites are shown below the qPCR primer pairs used to probe SSB enrichment (Fig. 5A). The cluster of *chi* sites located 1-2 kb upstream of the nick site correlates well with the observed peak enrichment of SSB in nicking strains (Fig. 5A). To further investigate the connection between *chi* site location and SSB enrichment, the endogenous *addA/addB* genes were deleted and functionally complemented by the homologous genes from *Staphylococcus aureus* (*rexA/rexB*, Supplementary Fig. 12E). While these enzymes are related, *B. subtilis* and *S. aureus chi* sequences are distinct (*chi*$^{Bs}$ is 5'-AGCGG-3' and *chi*$^{Sa}$ is 5'-GAAGCGG-3')[59,70,71] and accordingly the locations of the respective *chi* sites upstream of the targeted nick at -90° are different (Fig. 5A). Revaluation of SSB enrichment by ChIP in the strain expressing RexAB showed that the SSB profile is shifted upstream to a position proximal to the first *chi*$^{Sa}$ sites (Fig. 5A). Taken together, the data suggest that end-processing is necessary to deposit SSB onto ssDNA generated

from a seDSB, and that AddAB nuclease activity can be abolished without compromising recombinational repair.

## The SSB C-terminal tail is required for recombinational repair at a nick

SSB forms a homotetramer that binds ssDNA non-specifically via its N-terminal domain[72]. Each SSB monomer harbours a long disordered C-terminal domain that contains a protein interaction sequence at the terminus (the C-terminal tail or CTT) (Fig. 6A-B, Supplementary Fig. 13A)[73]. While the *B. subtilis* SSB-CTT is dispensable for cell viability, it is required for efficient repair of DNA lesions created by either UV irradiation or mitomycin C treatment[49,74].

To determine whether the SSB-CTT is required for recombinational repair following replisome inactivation at a nick, an IPTG-inducible SSB complementation system was generated in a Δ*ssbA* strain, allowing exogenous expression of either the wild-type protein or a variant lacking the last six amino acids (SSB$^{\Delta CTT}$, Supplementary Fig. 13B). Spot-titre analyses confirmed that SSB$^{\Delta CTT}$ can complement a Δ*ssbA* mutant similar to wild-type SSB and that viability was dependent upon IPTG (Supplementary Fig. 13C), while immunoblots showed that the inducible SSB proteins were being expressed at levels comparable to the endogenous protein (Supplementary Fig. 13D).

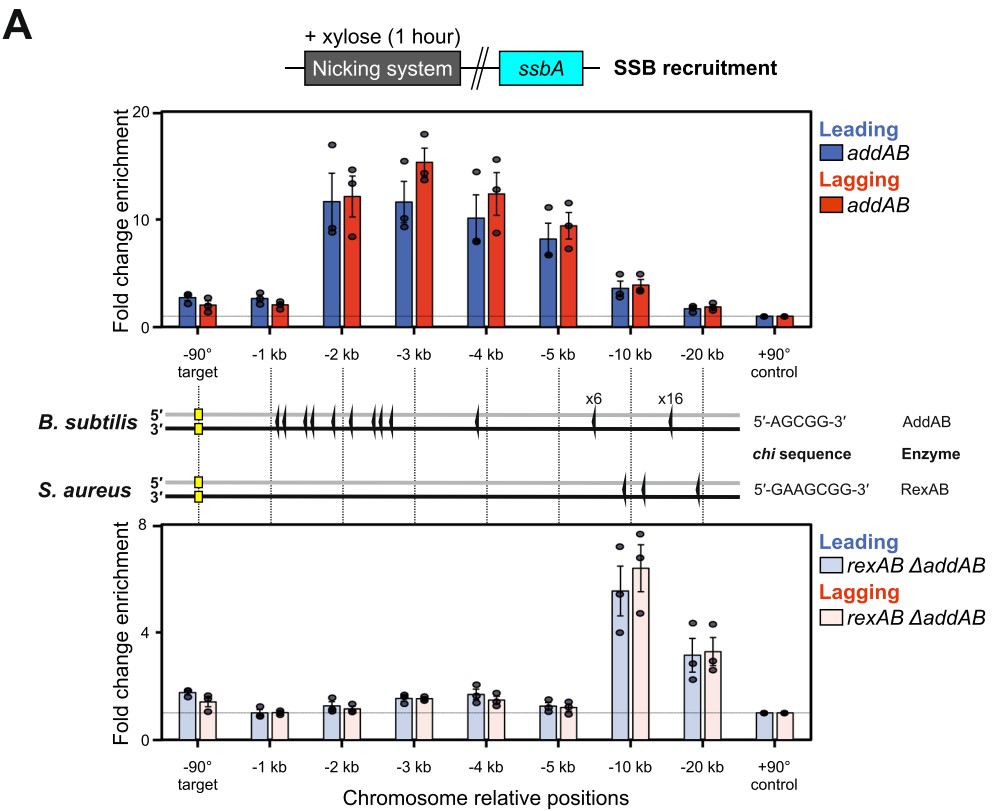

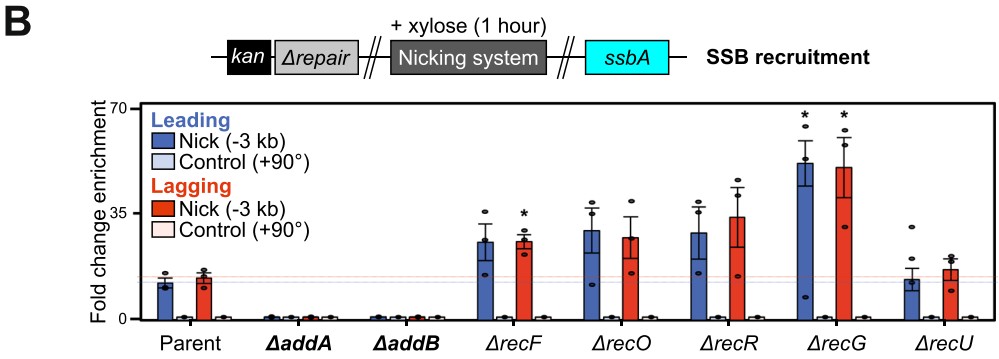

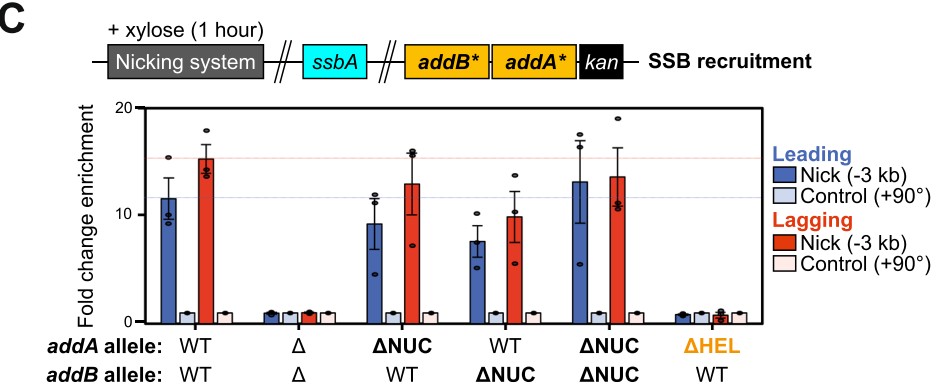

The SSB complementation system was then moved into strains capable of expressing nCas9 and sgRNAs (Supplementary Fig. 13E). Spot-titre analyses showed that wild-type SSB supports viability following induction of nicking, whereas the SSB$^{\Delta CTT}$ variant could not (Fig. 6C). MFA confirmed that chromosome replication from the nick site to the terminus is arrested when SSB lacks its C-terminal tail

(Fig. 6D). Therefore, the results indicate that the SSB-CTT is necessary for DNA repair following replisome inactivation at a nick.

**The SSB-CTT recruits RecO during recombinational repair**

The SSB-CTT acts as a protein interaction hub[49] that directly interacts with several factors shown above to be necessary for recombinational

**Fig. 5 | SSB requires AddA helicase activity for loading upstream of a nick.**
**A** ChIP-qPCR analyses of SSB in strains engineered to express nCas9 in the presence of either endogenous AddAB (top) or heterologous RexAB (bottom) end-processing machinery. Locations of respective *chi* sites are indicated. The sgRNAs targeted nCas9 to a locus located at -90° and protein enrichment was normalised to the control locus at +90°. **B** ChIP-qPCR analyses of SSB in strains with recombinational repair mutants engineered to express nCas9. qPCR primers amplified either a site located -3 kb from the sgRNA target (-90°) or a control locus (+90°) used for normalisation. Note a significant increase in SSB enrichment in some of mutants tested under these conditions. Two-sided Student t-test results: * indicates *p*

value = 0.0143 for lagging strand Δ*recF*, *p* value = 0.1666 and *p* value = 0.0223 for leading/strand Δ*recG* backgrounds, respectively (n = 3, see Methods/Source Data for details). **C** ChIP-qPCR analyses of SSB in strains with AddA and AddB variants engineered to express nCas9. Alleles refer to wild-type (WT), knockouts (Δ), and point mutations inactivating nuclease (ΔNUC) or helicase (ΔHEL) activities. qPCR primers amplified either a site located -3 kb from the sgRNA target (-90°) or a control locus (+90°) used for normalisation. **A**–**C** Error bars indicate the standard error of the mean and circles overlaid on bars correspond to three biological replicates. Source data are provided as a Source Data file.

repair, including RecO, RecG, and PriA (Figs. 2I–K, 3A, B, and Supplementary Fig. 13A). ChIP showed that the SSB^ΔCTT variant was enriched upstream of a nick at similar levels to the wild-type protein (Fig. 6E), indicating that the growth defect of the SSB^ΔCTT variant following nicking was not due to a general loss of SSB activity. To investigate the essential role of the SSB-CTT, ChIP was employed to probe the recruitment of His-RecO upstream of nick sites. Enrichment of His-RecO was abolished in cells harbouring SSB^ΔCTT (Fig. 6E), indicating that the SSB:RecO interaction depends on the SSB-CTT (Supplementary Fig. 13F).

The RecFOR system is thought to promote RecA binding ssDNA by stimulating the dissociation of SSB[23,75]. ChIP confirmed that enrichment of His-RecA was dependent upon both AddAB and RecFOR systems (Fig. 6F). Critically, enrichment of His-RecA was also dependent upon the SSB-CTT (Fig. 6E). Taken together, these results are consistent with the model that AddAB processes a seDSB to generate ssDNA for SSB, followed by RecFOR mediating displacement of SSB to allow RecA loading on ssDNA[23,75–77]. Further, they suggest that an essential role of the SSB-CTT is to recruit RecO for this process.

### RecG is recruited to sites of recombinational repair
ChIP enrichment of RecA was independent of RecG (Fig. 6F), suggesting that RecG acts downstream of RecA-mediated strand exchange, consistent with proposed activities in DNA remodelling. RecG has been implicated in a myriad of DNA transactions including reversal of replication forks, migration of Holliday junctions, regulation of RecA strand exchange, processing of flaps generated when DNA replication forks converge, and stabilisation of D-loops[65,78–81]. To determine whether RecG acts directly during recombinational repair, the endogenous *recG* allele was fused to a FLAG epitope tag and recruitment of FLAG-RecG was probed using ChIP following induction of nCas9 and sgRNAs in exponentially growing cells. The results showed that FLAG-RecG is maximally enriched ~3 kb upstream of a nick in either leading or lagging strand (Fig. 7A), consistent with RecG playing a direct role in DNA remodelling necessary to resume DNA replication (Fig. 3B)[49].

### RecU is recruited to sites of recombinational repair
Of the recombination proteins found to be important for supporting growth following nick induction, the Holliday junction resolvase RecU was conspicuous[28]. First, deletion of *recU* impaired growth less than the other mutants (Fig. 3A). Second, MFA indicated that DNA replication past a nick was independent of *recU* (Fig. 3B, Supplementary Fig. 11B). Therefore, we considered the possibility that growth defects associated with Δ*recU* following chromosome nicking were indirect, as the deletion mutant is known to elicit diverse chromosome segregation abnormalities (Fig. 7B, Supplementary Fig. 14)[82]. To address this hypothesis, the endogenous *recU* allele was fused to a FLAG epitope tag and enrichment of RecU-FLAG was determined using ChIP. The results show that RecU-FLAG was specifically recruited to DNA upstream of a nick site (Fig. 7E), consistent with a direct role in Holliday junction resolution. However, due to the pleiotropic phenotype of a Δ*recU* mutant, its precise importance during recombinational repair remains unclear.

## Discussion
In this study we have characterised the fate of the bacterial replisome in vivo after it encounters a single-strand discontinuity in template DNA (Fig. 1A). The results indicate that a nick in the template for either the leading or lagging strand arrests DNA synthesis, likely generates a seDSB that elicits repair by homologous recombination, and is followed by PriA-dependent replication restart (Figs. 1–3). These findings are consistent with the observation that a nick in a replicating bacteriophage chromosome generates DSBs[5] and with recent reports of the impact of nicks in eukaryotic replication systems[14,15,22]. Importantly, while replisomes are thought to be generally tolerant to many types of DNA damage[83], it appears that the bacterial replication machinery cannot readily bypass single-strand interruptions as these lesions affect helicase progression (Figs. 2, 8(I)).

Replication beyond a nick requires recombinational repair (Fig. 3). Genetic analysis combined with ChIP indicates that a single pathway is utilised under these experimental conditions: AddAB unwinds the DNA (Fig. 8(II)); SSB binds the ssDNA produced by AddAB (Fig. 8(III)); RecFOR counteracts SSB and allows RecA to bind ssDNA (Fig. 8(IV, V)); RecA promotes strand invasion of the homologous chromosome and generates a D-loop (Fig. 8(VI)); RecG remodels a DNA substrate formed during recombination (Fig. 8(VII)); PriA binds to the repaired replication fork (Fig. 8(VIII)) and reloads helicase to promote replisome assembly and DNA replication restart (Fig. 8(IX)). The observed lack of redundancy between proteins involved in recombinational DNA repair suggests that the pathway identified here is the major route for fixing broken replication forks in *B. subtilis*[48,84].

Building upon these findings, we investigated specific protein activities to help elucidate their roles during recombinational repair of a replisome inactivated at a single-strand discontinuity. Below we discuss these findings in what we propose is chronological order.

### The replicative helicase is inactivated after it encounters a nick
In bacteria the replicative helicase encircles and tracks along the template for lagging strand synthesis. ChIP data suggests that there are distinct fates of the helicase depending on the location of a nick (Fig. 2A, B, Supplementary Fig. 4). As expected for a nick in the lagging strand template, helicase enrichment was only observed at the cleavage site and upstream, consistent with the model that helicase runs-off the template when it encounters the single-strand discontinuity (Figs. 2G, 8(I)). In contrast, when the nick was in the leading strand template, helicase was also enriched downstream of the cleavage site, suggesting that the enzyme continued translocation. We propose that upon encountering a nick in the leading strand template, helicase slides onto the dsDNA template in an inactive conformation that cannot perform further DNA unwinding (Figs. 2G, 8(I))[40]. This model is consistent with in vitro experiments using *Thermus aquaticus* helicase, where after unwinding up to a leading strand nick, the enzyme can continue to translocate with the two strands of DNA passing through the central channel with no resultant unwinding[40]. Whether the bacterial helicase is actively removed from dsDNA under these conditions, akin to what has been observed for the eukaryotic CMG helicase in *Xenopus* extracts[14], requires further investigation.

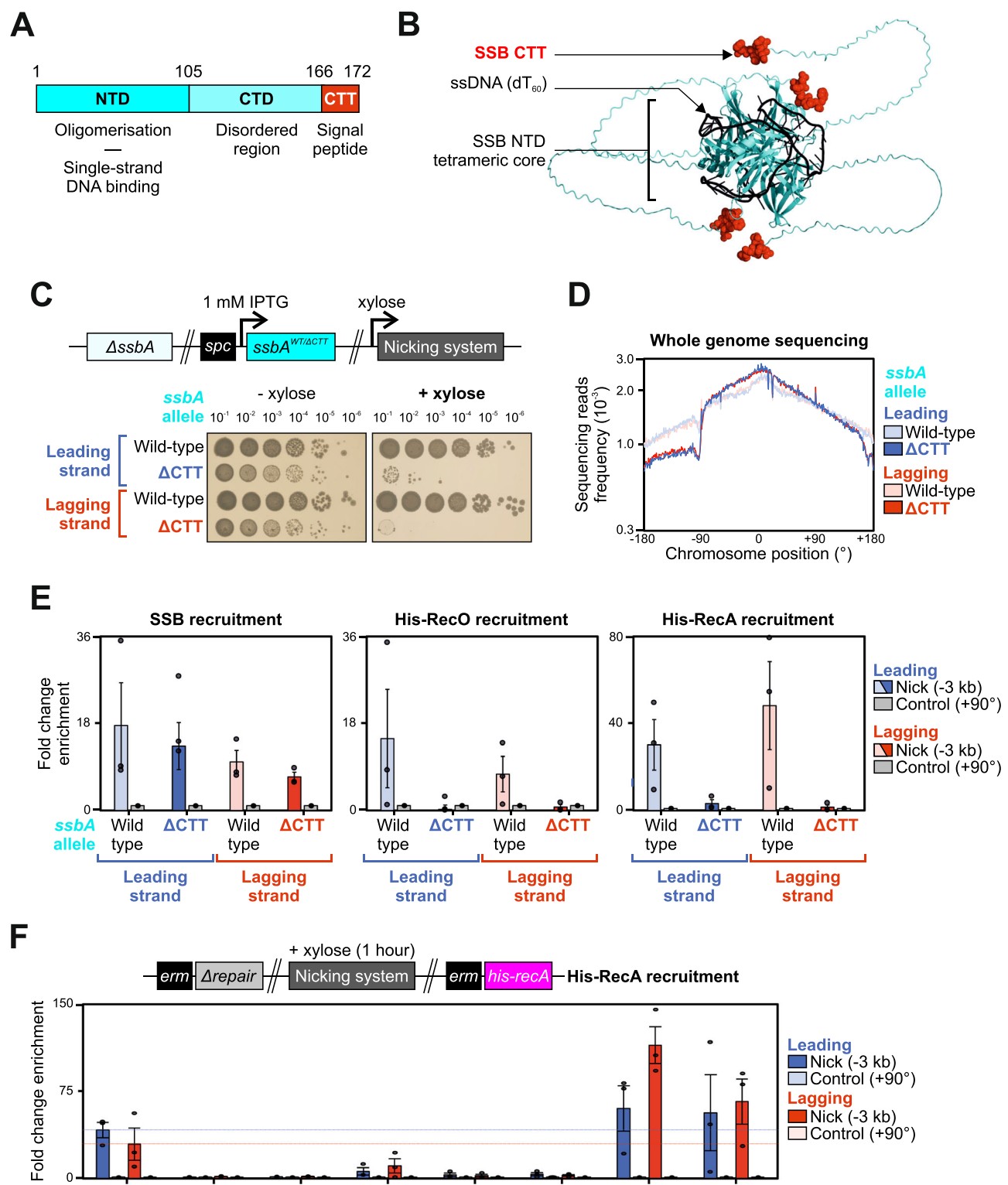

## AddAB helicase activity is necessary and sufficient for recombinational repair

AddAB and related enzymes have been shown to bind blunt end dsDNA or dsDNA with short overhangs (4-20 nucleotides with either a 5′- or a 3′-end)[59,85]. Moreover, AddAB is inert on DNA substrates lacking a blunt end[57]. The requirement for AddAB to efficiently process DNA past a nick (Fig. 3), specifically AddAB helicase activity to recruit SSB (Figs. 4C, D, 5), is consistent with the model that a seDSB is generated when a bacterial replisome encounters a single-strand discontinuity.

When the template for leading strand synthesis is nicked, the DNA polymerase likely runs-off the end of the substrate to produce a blunt or nearly blunt end, suitable for recognition by AddAB. However, when the template for lagging strand synthesis is nicked, a substrate with a 3′-tail would likely be produced. Because the reverse genetic screen showed that no other major nucleases are required for this recombinational repair pathway (Fig. 3, Supplementary Figs. 8, 12D and Supplementary Data 1), we speculate that the high local concentration of replication factors could facilitate priming near the end of the 3′-tail,

**Fig. 6 | The SSB-CTT is essential to recruit RecO following replisome inactivation at a single-strand discontinuity. A** Schematic of SSB primary structure with corresponding functions. Numbers indicate amino acid residue boundaries between the N-terminal domain (NTD), C-terminal domain (CTD) and C-terminal tail (CTT). **B** Structural model *of B. subtilis* SSB homotetramer in the presence of single-strand DNA (AlphaFold 3). A sixty nucleotide homopolymer of thymidine bases ($dT_{60}$) is shown in black and SSB-CTT residues are highlighted in red. **C** Spot-titre analysis of SSB variants in strains engineered to express (+ xylose) nCas9. Top panel shows the genetic system employed in this experiment. sgRNAs were targeted to a locus at -90°. **D** MFA using whole genome sequencing in strains with SSB variants engineered to express nCas9. The frequency of sequencing reads was plotted against genome position. **E** ChIP-qPCR analyses of SSB, His-RecO and His-RecA proteins in strains engineered to express nCas9. qPCR primers amplified either a site located -3 kb from the sgRNA target (-90°) or a control locus (+90°) used for normalisation. Error bars indicate the standard error of the mean and circles overlaid on bars correspond to three biological replicates. **F** ChIP-qPCR analyses of His-RecA in strains with recombinational repair mutants engineered to express nCas9. Top panel shows the genetic system employed in this experiment. qPCR primers amplified either a site located -3 kb from the sgRNA target (-90°) or a control locus (+90°) used for normalisation. Error bars indicate the standard error of the mean and circles overlaid on bars correspond to three biological replicates. Source data are provided as a Source Data file.

thereby filling the ssDNA with an Okazaki fragment to create a suitable substrate for AddAB (Fig. 8(I-II)). Alternatively, unidentified nucleases may degrade the ssDNA tail[86].

The master recombination factor RecA requires ssDNA to promote homologous pairing. A long-standing question is how this ssDNA is generated in vivo. Most current models suggest that enzymes resect a DSB to generate a single-strand terminating with a 3′-end[28,34]. Here we tested this model by mutating the nuclease activities of AddA and AddB. The results showed that neither nuclease activity is required for recombinational repair under the experimental conditions tested (Fig. 4C, D). Moreover, deletion of the major exonuclease RecJ in a strain expressing the nuclease-dead AddAB variant did not display a synthetic phenotype (Supplementary Fig. 12D). In contrast, AddAB helicase activity was required for recombinational repair (Fig. 4D), consistent with a previous study of the ancestral AdnAB end-processing machinery in *Mycobacteria tuberculosis*[87] and supported by *chi*-dependence to facilitate SSB loading (Fig. 5A, Supplementary Fig. 12E). The local concentration of SSB is high near the site of DNA replication[88,89], which together with phase separation[90,91] may facilitate its rapid association with ssDNA emerging from AddAB. In fact, in vitro the AddA$^{\Delta NUC}$AddB$^{\Delta NUC}$ nuclease-dead enzyme does not unwind dsDNA in the absence of SSB[92], suggesting that AddAB unwinding and SSB binding ssDNA may be coupled. Together these results raise the unexpected possibility that during recombinational repair at a nick, the key activity during end-processing is to separate the strands and create a ssDNA substrate for SSB (Figs. 5C, 8(II-III)).

### Interaction of the SSB C-terminal tail with RecO is necessary for recombinational repair

The SSB-CTT is an established protein interaction hub that has been shown to directly bind several of the essential replication and repair factors required at nicks, including RecO[93–96], RecG[97,98], and PriA[74,99] (Supplementary Fig. 13A). However, the critical activities performed by the SSB-CTT in vivo had not been established[49]. Based on the results above, we conclude that the SSB-CTT interaction with RecO is necessary for recombinational repair (Fig. 6). RecO is part of the RecFOR system required for RecA loading onto ssDNA in vitro[76] and RecA-GFP foci formation in vivo[23]. Because RecO activity precedes the activities of RecG and PriA (i.e., RecA binding ssDNA is blocked, Fig. 6F), further conclusions regarding the importance of these protein:protein interactions cannot be drawn.

Historically the RecFOR system has been proposed to mediate ssDNA gap-filling, rather than recombination at a DSB[61]. While it is possible that multiple types of DNA lesions are associated with nicks generated by nCas9 in *B. subtilis*, we note that recruitment of SSB and RecA was dependent on AddAB (Figs. 5B, 6F). Because AddAB-dependent end-processing is not associated with models for recombinational repair of ssDNA gaps, the data presented here is most consistent with RecFOR acting within the proposed seDSB repair pathway (Fig. 8). Supporting this viewpoint, AddAB has been shown to be epistatic to RecFOR in *Thermus thermophilus*[100] and RecFOR is essential for DSB repair in *Deinococcus radiodurans*[101].

In vitro the RecO:RecR complex facilitates loading of RecA onto SSB-coated ssDNA[102,103]. Because RecO, RecR, and RecF were all required for recombinational repair following nicking (Fig. 3), we propose that the essential role of RecF here is to recruit RecR and generate the RecO:RecR complex. This model is consistent with a hand-off mechanism for RecR observed in vitro[102]. Since SSB and RecO are enriched at a broken replication fork (Fig. 6E), we predict that there is a mechanism (potentially facilitated via RecF) to regulate RecO:RecR complex formation until it is required, thus safeguarding the SSB-coated DNA from promiscuous RecA loading (Figs. 5B, 6F and 8(III, V)).

### RecG is necessary to complete recombinational repair prior to replication restart

The biological role of the DNA translocase RecG has been a long-standing question[64,65]. MFA shows that a Δ*recG* mutant cannot efficiently replicate past a nick (Fig. 3B) and ChIP indicates that RecG is enriched upstream of the lesion (Fig. 7A)[54]. We propose that RecG acts to remodel a DNA substrate formed after RecA-mediated strand invasion, either the Holliday junction or the replication fork (Fig. 8(VI, VII)). Branch migration of the Holliday junction away from the replication fork could be required for downstream steps necessary for PriA-dependent helicase loading[104]. Alternatively, RecG may process the replication fork itself, preparing it for PriA loading on the correct substrate (Supplementary Fig. 15)[54,55].

### Holliday junction migration and resolution are not necessary for replication restart

While *recU* was identified as important in our genetic screen following nicking (Fig. 3A) and RecU was specifically enriched upstream of a nick (Fig. 7C), it was found that RecU is not necessary for replication restart per se (Fig. 3B). Single cell analyses revealed significant chromosome segregation defects in all Δ*recU* mutants (Fig. 7B, Supplementary Fig. 14)[82], likely contributing to the observed loss in viability (Fig. 3A, Supplementary Fig. 7A). Together these results support the model that recombinational repair is mechanistically distinct from resolution of conjoined chromosomes[56].

### Potentiating ROS-mediated antibiotic toxicity by perturbing DNA repair

In this work, the nicks generated by Cas9 enzymes were used as proxies for single-strand discontinuities formed during removal of damaged DNA from a bacterial genome. We find that when a bacterial replication fork encounters a single strand discontinuity it becomes inactivated. By identifying the essential factors and activities required to repair the DSB formed after a replisome encounters a nick, we have provided a set of molecular targets whose disruption could potentiate the activity of antimicrobial compounds that damage DNA.

## Methods
### Experimental model and growth
The following bacterial organisms were used: *Escherichia coli* and *B. subtilis*. Unless otherwise stated in method details, strains were grown

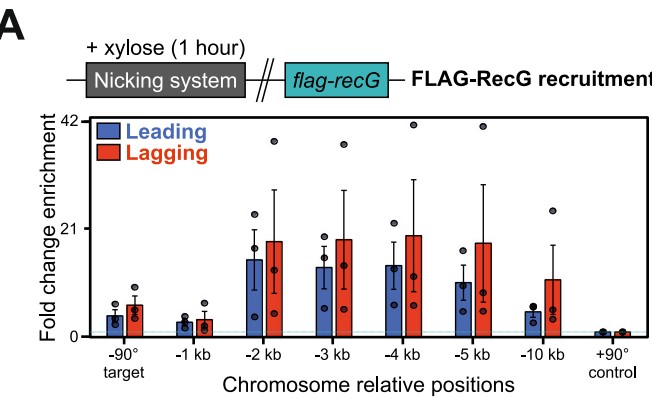

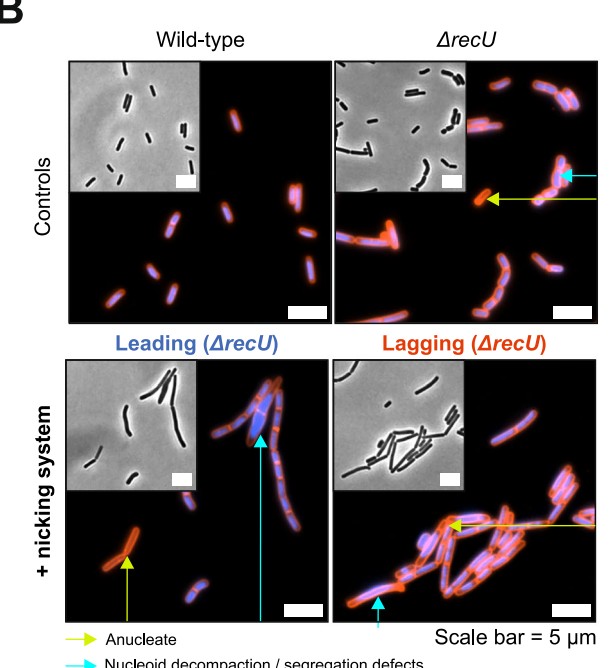

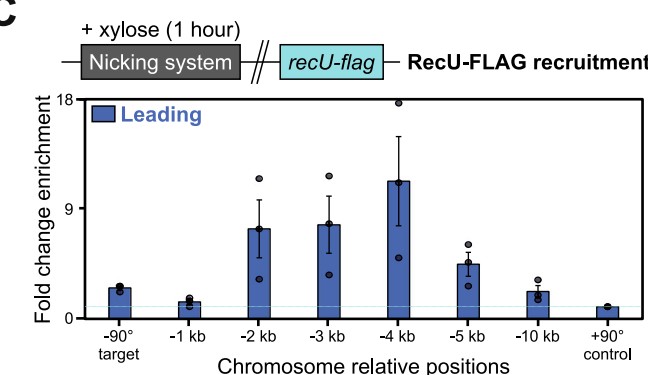

**Fig. 7 | RecG and RecU are recruited to sites of recombinational repair. A** ChIP-qPCR analyses of FLAG-RecG in strains engineered to express nCas9. Top panel shows the genetic system employed in this experiment. The sgRNAs targeted nCas9 to a locus located at -90° and protein enrichment was normalised to the control locus at +90°. Error bars indicate the standard error of the mean and circles overlaid on bars correspond to three biological replicates. **B** Fluorescence microscopy images of Δ*recU* mutant strains engineered to express nCas9 (sgRNAs targeted to -90°). Grey scale images correspond to phase contrast. Blue signal corresponds to DAPI staining (DNA) and red signal corresponds to Nile red staining (membrane) in composite fluorescence images. Cas9 proteins were induced (+ xylose) during exponential growth phase for 90 minutes before cells were collected for imaging. Scale bar indicates 5 μm. **C** ChIP-qPCR analyses of RecU-FLAG in a strain engineered to express nCas9. Top panel shows the genetic system employed in this experiment. The sgRNA targeted nCas9 to a locus located at -90° and protein enrichment was normalised to the control locus at +90°. Error bars indicate the standard error of the mean and circles overlaid on bars correspond to three biological replicates. Source data are provided as a Source Data file.

## Oligonucleotides

Oligonucleotides were purchased from Eurogentec and Sigma to assemble recombinant DNA (sequences are listed in Supplementary Data 3) or perform quantitative PCR (Supplementary Data 4). Design of sgRNA targets was performed using the online portal CRISPick with default parameters[105] and recombinant DNA assembly was achieved via in vitro methods (see below).

## *E. coli* plasmid construction

*E. coli* transformations for constructs harbouring *priA* were performed in CW198 via heat-shock following the Hanahan method[106] and propagated in LB with appropriate antibiotics at 37 °C. DH5α [F-Φ80lacZΔM15 Δ(lacZYA-argF) U169 recA1 endA1 hsdR17(rk-, mk + ) phoA supE44 thi-1 gyrA96 relA1 λ-] was used for other plasmids construction. Plasmids were purified using QIAprep spin miniprep kits and recombinant DNA confirmed by sequencing (Supplementary Data 3). Descriptions, where necessary, are provided below.

pCW367, pCW368, pCW478, pCW479, pCW510, pCW717, pCW734, pCW736 and pCW850 were generated by Quickchange mutagenesis using mutagenic primer pairs containing an overlap of 10-15 bp. PCR products were treated with DpnI for at least 3 hours to digest template DNA and 10 μl were used for *E. coli* transformation.

pCW20, pCW300, pCW301, pCW327, pCW460, pCW471, pCW472, pCW575, pCW734, pCW736 and pCW766 were generated by Gibson assembly using the NEB Hi-Fi cloning kit according to manufacturer instructions following purification of individual PCR fragments using QIAquick PCR purification kits.

## Linear recombinant DNA assembly

For recombinant DNA assembly without *E. coli* propagation, pCW573, pCW574, pCW577, pCW578, pCW605, pCW640, pCW672, pCW702, pCW720, pCW732, pCW758, pCW759, pCW760, pCW777, pCW779, pCW781, pCW783, pCW785, pCW802, pCW803, pCW804, pCW809, pCW834, pCW848, pCW854, pCW855, pCW857 and pCW859 were assembled linearly using a multistep in vitro assembly process followed by nested PCR amplification prior to *B. subtilis* transformation. Briefly, individual PCR fragments corresponding to homology regions and insert DNA were amplified, purified and DNA ends ligated using the NEB Hi-Fi cloning kit. Raw Gibson assembly products were then amplified using nested primer pairs annealing towards the end of homology regions and PCR products were used for *B. subtilis* transformation. Following isolation of *B. subtilis* colonies resistant to the appropriate antibiotic markers, genomic DNA was extracted using the Qiagen DNeasy blood and tissue kit, regions corresponding to recombinant DNA were amplified via PCR using oligonucleotides annealing outside homology fragments to ensure double crossover

at 37 °C using nutrient agar (NA) or lysogeny broth (LB: 10 g/L tryptone, 10 g/L NaCl, 5 g/L yeast extract) for routine selection and maintenance of bacterial strains. Supplements were added as required: ampicillin (100 μg/ml), erythromycin (1 μg/ml) in conjunction with lincomycin (25 μg/ml), chloramphenicol (5 μg/ml), kanamycin (5 μg/ml), spectinomycin (50 μg/ml), zeocin (10 μg/ml), xylose (1% w/v) and IPTG (0.05 mM, 0.1 mM or 1 mM).

## Bacterial strains

Bacterial strains used in this study are listed in Supplementary Data 2.

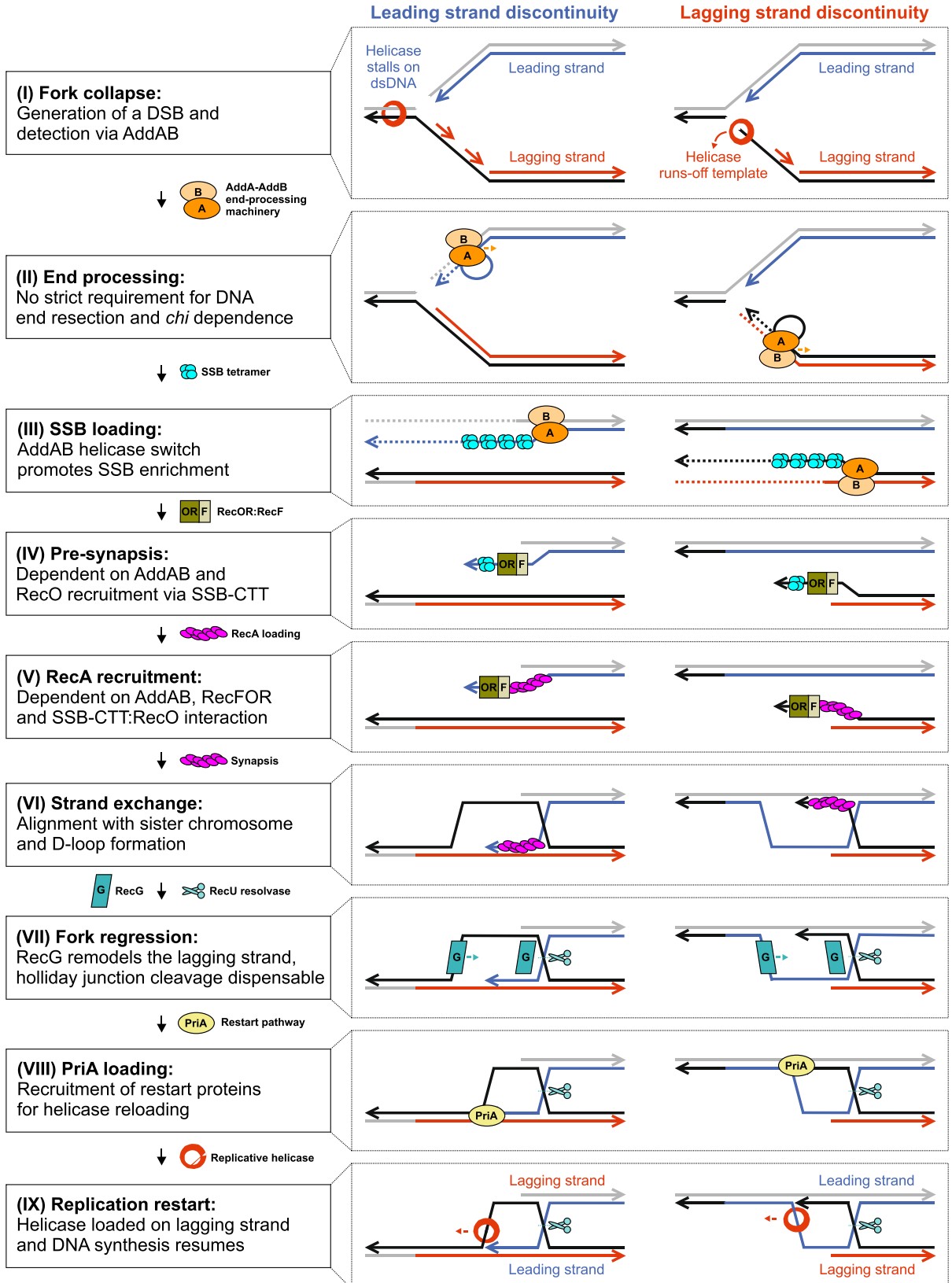

**Fig. 8 | Model for seDSB repair and helicase reloading following bacterial replisome inactivation at a single-strand discontinuity.** Model illustrating molecular events that are required or dispensable for recombinational repair of a seDSB created after a replication fork encounters a single-strand discontinuity located in either the leading or lagging strand template.

had occurred and results were confirmed via PCR product sequencing. Note that CW2045 and derivatives harbour the additional $addA^{D682E}$ mutation necessary to propagate $addB^{D961A}$-$addA^{D1172A}$ in live cells, which is surface exposed and structurally unlikely to impact AddAB end-processing activities (Supplementary Fig. 12F).

## *B. subtilis* strain construction

Transformation of competent *B. subtilis* cells was performed using an optimised two-step starvation procedure[107]. Briefly, recipient strains were grown overnight at 37 °C in transformation medium (Spizizen salts: 0.2% w/v ammonium sulphate, 1.4% w/v dipotassium phosphate, 0.6% w/v potassium phosphate, 0.1% w/v tri-sodium citrate, 0.02% w/v magnesium sulphate supplemented with 1 µg/ml ammonium iron(III) citrate, 6 mM MgSO4, 0.5% w/v glucose, 0.02 mg/ml tryptophan and 0.02% w/v casein peptone) supplemented with IPTG where required. Overnight cultures were diluted 1:20 into fresh transformation medium supplemented with IPTG where required and grown at 37 °C for 2.5-3 h with continual shaking. An equal volume of prewarmed starvation medium (Spizizen salts supplemented with 6 mM MgSO4 and 0.5% w/v glucose) was added and the culture was incubated at 37 °C for two hours with continual shaking. DNA was added to 350 µl cells and the mixture was incubated at 37 °C for one hour with continual shaking. 20-200 µl of each transformation was plated onto selective media supplemented with IPTG where required and incubated at 37 °C for 24-48 h. Results were validated by propagating cells on relevant antibiotic markers and sequencing. Sanger sequencing identified that the strong nCas9/sgRNA nicking system is prone to accumulation of suppressor mutations either disrupting protospacer or its promoter sequence (Supplementary Fig. 1F), therefore it is recommended to rebuild these strains periodically to validate their identity.

For strain construction of essential gene knockouts (e.g., *ssbA* and *priA*), an ectopic copy of the gene of interest was placed under the control of an IPTG-inducible promoter and recombinant products were transformed in wild-type *B. subtilis*. Note that *priA* was C-terminally fused to a degradation tag (*priA-ssrA*) to promote proteolysis and reduce background expression in the absence of IPTG. Cells harbouring an ectopic expression cassette were transformed with recombinant DNA to knockout the corresponding endogenous gene copy and colonies were obtained in the presence of IPTG to avoid accumulation of suppressor mutations. Thereafter, these strains were always propagated in the presence of IPTG to enable genetic complementation via the inducible essential gene copy.

For the construction of protein fusions, *ssrA* tags (AANDENYSENYALAA) were directly fused to C-terminal ends of target proteins (e.g., Cas9-ssrA variants and PriA-ssrA), whereas the fluorescent mNG-RecA DNA break reporter was constructed by N-terminal fusion via a short serine/glycine linker (SG3).

## Spot-titre assays

Cells were grown in LB overnight in the presence of 0.5% w/v glucose. Overnight cultures were diluted in fresh LB medium using 10-fold serial dilutions and 5 µL aliquots were spotted onto NA plates with or without xylose. NA plates were supplemented with IPTG when required as indicated on individual figure panels. All plates were incubated for 24 h or up to 48 h when indicated. Experiments were performed independently at least three times and representative data are shown.

## Antibody purification

SSB anti-serum was used as delivered from Eurogentec. Other primary antibodies received from Eurogentec (anti-DnaI, anti-DnaC, anti-PriA) were affinity purified prior to use. Anti-sera were diluted 1:1 with PBS and applied to respective antibody/NHS-activated agarose columns. Purified antibodies were eluted with glycine, dialysed with PBS and

concentrated. Affinity purified antibodies were stored at -20 °C in 1X PBS with 50 % v/v glycerol. The antibody validation report is provided with the Source Data.

## Immunoblot analyses

Cultures were grown overnight in LB and diluted 1:100 in fresh medium the next morning. Diluted cells were grown to an absorbance at 600 nm ($A_{600}$) of 0.2-0.3, induced with 1% w/v xylose or 1 mM IPTG where required and further incubated for one hour. Cells were harvested using centrifugation, resuspended in 1X PBS supplemented with one tenth of a peptidase inhibitor tablet and sonicated (40 amp) twice for 12 seconds with 3 second pulses at 4 °C. Samples were adjusted to 1X LDS buffer and 125 mM DTT, fixed for 5 minutes at 95 °C and centrifuged for 2 minutes prior to loading the supernatant on polyacrylamide gels. Proteins were separated by electrophoresis using NuPAGE 4-12% Bis-Tris gradient gels run in MES SDS buffer and transferred to methanol-activated Hybond-P PVDF membranes using Wypall X60 cloths for blotting in a semi-dry apparatus (Bio-rad Trans-Blot Turbo, transfer buffer: 300 mM Tris-HCl, 300 mM glycine, 140 mM tricine, 0.05% w/v SDS, 2.5 mM EDTA). Membranes were washed twice using PBST (1X PBS supplemented with 0.1% v/v Tween-20) prior to blocking for 90 minutes at room temperature (blocking buffer: PBST supplemented with 7.5% w/v milk). Individual polypeptides were probed using the following primary antibodies diluted in blocking buffer following overnight incubation at 4 °C: anti-Cas9 (1:1000, Merck SAB4200701), anti-FtsZ[108] (1:5000), anti-PriA (1:1000, Eurogentec), anti-SSB (1:1000, Eurogentec) and anti-mNeonGreen (1:1000, ProteinTech 32F6). Detection was performed with anti-rabbit (1:10000 for PriA/SSB, ProteinTech SA00001-2), anti-sheep (1:5000 for FtsZ, Merck A3415) or anti-mouse (1:5000 for Cas9/mNeonGreen, ProteinTech SA00001-1-A) horseradish peroxidase-linked secondary antibodies in combination with the Clarity ECL substrate using an ImageQuant LAS 4000 mini digital imaging system. Detection of Cas9, FtsZ, PriA, SSB and mNeonGreen was within a linear range. Experiments were independently performed at least twice and representative data are shown.

## Whole genome sequencing

Cells were grown in rich chemically defined medium (RCDM: Spizizen salts supplemented with 1 µg/ml ammonium iron(III) citrate, 0.1 mM CaCl2, 0.13 mM MnSO4, 6 mM MgSO4, 0.5% w/v glucose, 0.1% w/v glutamate, 0.02 mg/ml tryptophan and 0.02% w/v casein peptone) overnight and diluted 1:100 in fresh RCDM the next morning. Diluted cells were grown to an absorbance at 600 nm ($A_{600}$) of 0.2-0.3, induced with 1% w/v xylose and further incubated for one hour. Growth was arrested by adding 0.05% w/v sodium azide to cultures followed by vigorous mixing, cells were collected by centrifugation, the supernatant discarded, and pellets were flash frozen in liquid nitrogen before genomic DNA extraction using the Qiagen DNeasy blood and tissue kit and a final elution volume of 50 µl in ultrapure water. Individual Illumina gDNA libraries were prepared using the DNA Prep tagmentation kit and pre-paired i5/i7 indices prior to pooling, and DNA quantification was performed with the 1X dsDNA high sensitivity kit using a Qubit instrument. Pooled libraries were diluted to 100 pM in resuspension buffer and 20 µl were loaded onto iSeq 100 i1 Reagent v2 sequencing chips for whole-genome sequencing using an iSeq100 instrument.

Following whole-genome sequencing and demultiplexing embedded within the iSeq100 indexed sequencing workflow, .FASTQ files were extracted and loaded onto the CLC Genomics Workbench (version 23) for initial data processing. Paired .FASTQ files were mapped onto the *B. subtilis* Δ6 reference genome using the 'Map Reads to Reference' function and a .BAM file was created from mapped sequencing reads for each sequenced genome. An in-house script

written in R (version 4.5) using the Rsamtools library was employed to deconvolute .BAM files to validate even read distribution for sense/antisense strands prior to calculating density coverage and extracting cumulated densities per chromosome position.

## Fluorescence microscopy

Strains were grown overnight at 37 °C in RCDM and the following day, cultures were diluted 1:100 into fresh imaging medium (RCDM where glucose is replaced by 0.5% v/v glycerol) in the presence of 0.1 mM IPTG for induction of mNeonGreen-RecA expression where required. Cells were allowed to grow to an $A_{600}$ of 0.3, induced with 1% w/v xylose and further incubated for 90 minutes. DAPI (5 μg/ml) and Nile red (1 μg/ml) stains were used to visualise the nucleoid and cell membrane following a 10-minute incubation on a benchtop incubator prior to imaging.

Cells were mounted on ~1.25% agar pads (in sterile ultrapure water) and a 0.13-0.17 mm glass coverslip was placed on top. Microscopy was performed on an inverted epifluorescence microscope (Nikon Ti) fitted with a Plan Apochromat Objective 100x/1.40 NA Oil Ph3. Light was transmitted from a CoolLED pE-300 white light source through a Sutter Instruments liquid light guide and images were collected using a Photometrics Prime camera. Chroma fluorescence filter sets were used with 100 ms GFP, 250 ms DAPI and 50 ms mCherry exposure times at 100% LED power. Digital images were acquired using NIS Elements (version 5) and analysed via the Fiji software (version 1.53)[109]. Quantification of mNeonGreen-RecA features was manually curated from the count of 100 individual cells. All experiments were independently performed at least twice and representative data are shown.

## Chromatin immunoprecipitation

Chromatin immunoprecipitation (ChIP) was performed by following an established protocol with minor modifications[35]. Strains were grown overnight at 37 °C in RCDM and the following day, cultures were diluted 1:100 into fresh medium and allowed to grow to an $A_{600}$ of 0.2-0.3. Cells were induced with 1% w/v xylose and further incubated for one hour prior to ChIP. Samples were adjusted to 1X PBS and crosslinked with 1% v/v formaldehyde for 10 minutes on a roller at room temperature, then quenched with 0.5 M glycine. Cells were pelleted at 4 °C, washed three times with ice-cold 1X PBS (pH 7.3), frozen in liquid nitrogen and stored at -80 °C. Frozen cell pellets were resuspended in 500 μl of lysis buffer (50 mM NaCl, 10 mM Tris-HCl pH 8.0, 20% w/v sucrose, 10 mM EDTA, 100 μg/ml RNase A, one quarter of a peptidase inhibitor tablet, 4 mg/ml lysozyme) and incubated at 37 °C for 30 min to degrade the cell wall. Protoplasts were supplemented with 500 μl of immunoprecipitation buffer (300 mM NaCl, 100 mM Tris-HCl pH 7.0, 2% v/v Triton X-100, one quarter of a peptidase inhibitor tablet) to lyse cells and the mixture was incubated at 37 °C for a further 10 minutes before cooling on ice for 5 minutes. Lysis and immunoprecipitation buffer volumes were multiplied by the number of antibodies to probe per sample. DNA samples were sonicated (40 amp) three times for 2 minutes at 4 °C to obtain an average fragment size of 500 to 1000 base pairs. Cell debris were removed by centrifugation at 4 °C and the supernatant transferred to a fresh Eppendorf tube. To determine the relative amount of DNA immunoprecipitated compared to the total amount of DNA, 100 μl of supernatant was removed, treated with 0.5 mg/ml pronase at 37 °C for four hours then stored on ice. To immunoprecipate protein-DNA complexes, 800 μl of the remaining supernatant was incubated with individual antibodies (2 μl anti-SSB, 2 μl anti-PriA, 2 μl anti-DnaD[36], 2 μl anti-DnaB[36], 2 μl anti-DnaI, 2 μl anti-DnaC, 2 μl anti-mNeonGreen (ProteinTech 32F6), 4 μl anti-His (Qiagen 34660) or 25 μl anti-FLAG M2 beads (Merck M8823) for 90 minutes at room temperature (or 120 min for FLAG-tagged strains). 750 μg of protein G Dynabeads (Invitrogen 10009D) were equilibrated by washing with bead buffer (1X PBS, 0.01% v/v Tween 20), resuspended in 25 μl of bead buffer and incubated with the sample supernatant for one hour at room temperature. Immunoprecipitated complexes were collected by applying the mixture to a magnet and washed with the following buffers in the respective order: 0.5X immunoprecipitation buffer for 15 min, 0.5X immunoprecipitation buffer supplemented with 500 mM NaCl for 15 min, stringent wash buffer (250 mM LiCl, 10 mM Tris-HCl pH 8.0, 0.5% v/v Igepal, 0.5% w/v sodium deoxycholate, 10 mM EDTA) for 20 min. Finally, protein-DNA complexes were washed a further three times with TET buffer (10 mM Tris-HCl pH 8.0, 1 mM EDTA, 0.01% v/v Tween 20) and resuspended in 100 μl of TEN buffer (10 mM Tris-HCl pH 8.0, 1 mM EDTA, 300 mM NaCl). Formaldehyde crosslinks of both the immunoprecipitate and total DNA were reversed by incubation at 65 °C for 18 h in the presence of 1 mg/ml proteinase K. DNA was then removed from the magnetic beads, cleaned using QIAquick PCR purification columns and used for quantitative PCR analyses.

## Quantitative PCR

Quantitative PCR (qPCR) was performed using the Luna qPCR master mix to measure relative amounts of DNA bound by SSB, PriA, DnaD, DnaB, DnaI, DnaC, mNeonGreen-AddA, mNeonGreen-AddB, His-RecO, His-RecA, FLAG-RecG and RecU-FLAG at specific genomic locations (e.g., near the nick site located at -90° on the chromosome or the origin *oriC* compared to a control site located at +90° on the chromosome). All PCR reactions were assembled using the QIAgility robotic workstation in 20 μl reaction volumes in a Rotor-Disc 100 and qPCRs were run on Rotor-Gene Q instruments. Standard curves were obtained using the Rotor-Gene Q software (version 2) to calculate the efficiency of each primer pair, which varied ~5% between sets. Oligonucleotides were designed to amplify specific genomic regions using the Primer3Plus tool[110], were typically 20-25 bases in length (Supplementary Data 4) and amplified a ~100 bp PCR product. Individual fold enrichment ratios were obtained as follows: first, every Ct value was converted to $1/2^{Ct}$ and technical triplicates were averaged to generate a single enrichment value; second, genomic location specific enrichment was normalised by corresponding values obtained at +90° on the chromosome. Error bars indicate the standard error of the mean for three biological replicates. Unpaired, two-sided Student t-tests were employed to evaluate enrichment significance where relevant for $n = 3$ biological replicates (each averaged from three technical replicates) with control/test groups corresponding to parent/mutant genetic backgrounds, respectively.

## Protein structure prediction and representation

Protein models for SSB:ssDNA and SSB:RecO were generated using AlphaFold 3 and polypeptides corresponding to individual protein entities as input[111]. Main and alternative models were manually examined and Model 0 was chosen to highlight key features using the Pymol Molecular Graphics software (version 3.1)[112]. Structural highlights of the AddAB complex were derived from the crystal structure PDB: 4CEH[68].

## Reporting summary

Further information on research design is available in the Nature Portfolio Reporting Summary linked to this article.

# Data availability

The SSB:ssDNA and SSB:RecO models generated by AlphaFold 3 in this study have been deposited in Zenodo (https://zenodo.org/records/14793919) and in ModelArchive (https://modelarchive.org/) with the accession codes ma-tnuut (https://modelarchive.org/doi/10.5452/ma-tnuut) and ma-2k00a (https://modelarchive.org/doi/10.5452/ma-2k00a). Raw sequencing data has been deposited to the European Nucleotide Archive (ENA) (https://www.ebi.ac.uk/ena/browser/home) with the accession code PRJEB102268 (https://www.ebi.ac.uk/ena/browser/text-search?query=PRJEB102268). For additional information and requests, please contact the corresponding authors. Source data are available on Figshare (https://doi.org/10.6084/m9.figshare.30118162) and are provided with this paper.

## Code availability

The R code used to process next-generation sequencing data used in this study has been deposited in Zenodo (https://zenodo.org/records/14793122)[113].

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

## Acknowledgements
This work was supported by a Wellcome Trust Early-Career Award [226338/Z/22/Z] to C.W. and Wellcome Trust Discovery Award [225811/Z/22/Z] to H.M. We would also like to express our gratitude to Frances Davison and Amandeep Kaur for technical assistance, James Grimshaw for microscope maintenance and Frederic Schramm for insightful discussions.

## Author contributions
C.W. and H.M. contributed to the conception/design of the work. C.W., K.J.S. and S.F. generated results presented in the manuscript. C.W. created figures. C.W. and H.M. wrote/edited the manuscript.

## Competing interests
The authors declare no competing interests.
