## [Transparent Peer Review file · Nature Communications]

Rescuing the bacterial replisome at a nick requires recombinational repair and helicase reloading

Corresponding Author: Dr Charles Winterhalter

Version 0:

Reviewer comments:

Reviewer #1

(Remarks to the Author)

In the manuscript entitled "Rescuing bacterial genome replication: essential functions to repair a double-strand break and restart DNA synthesis", Winterhalter and colleagues use a clever genetic system based on a modified Cas9 enzyme to induce DNA nicks in a site-specific manner. Such a nick mimics the state of DNA after cleavage of a damaged base (i.e. 8-oxoguanine). It therefore opens up the possibility of performing a mechanistic analysis of the repair pathways involved, particularly concerning how such a nick is repaired when converted into a DNA double-strand break when a replication fork arrives.

The authors utilise the ability to modify the cleavage strand of Cas9 to produce a nick on either the leading or lagging strand, an elegant strategy that further enables them to delve into the details of the repair mechanisms. They use multiple genetic strategies (deletion and separation of function mutants, degron tags, inducible complementation of essential genes, etc) combined with genome-wide Marker Frequency Analysis and Chip qPCR to provide a very comprehensive picture of the pathways involved. As far as I can tell, this is the first time that site-specific nicks have been analysed in live bacteria, and the work presented here is novel and important. The manuscript is well written and the figures clear and well presented. There are however some questions that should be clarified and are described below.

Major comments:

1)The introduction focuses on the base damage caused by ROS due to antibiotics. However, there are many other possibilities to create damage (indeed, the graphical abstract mentioned the immune system) and these should be mentioned too. Moreover, a brief mention of Base excision repair (which might be expected to be involved in the repair of the nick produced by Cas9?) would be helpful.

2)Another important point is to clarify whether the authors expect that the only type of DNA damage caused by their system is a replication-dependent double-strand break (DSB) or whether other types of damage can occur? In the first paragraph of the results section, the authors provide compelling evidence through MFA that replication is impaired by the nicks and this will definitely result in DSBs. However, the evidence provided does not prove that the only type of damage is a DSB. Localisation of RecA and formation of spots and bundles (from line 124) does not by itself prove that the damage is a DSB because RecA catalyses homologous recombination from DSB but also from single-strand gaps (RecFOR pathway). As far as I understand, the system creates a nick and not a single-nucleotide gap (i.e. the DNA backbone is cut but the nucleotides are still all present), but this is not clearly stated. Moreover, it is conceivable that the nick could lead to degradation of a few nucleotides on the nicked strand that would create a gap. Similarly, as shown by the authors on the diagram of figure 1A (right panel), even after replication, the strand that does not have a DSB shows a gap. Has the possibility of gap formation been completely ruled out?

In this case, one could conceivably observe repair by homologous recombination through the RecFOR gap repair pathway independently of replication (for example one can conceive that a gap is repaired before the replication fork arrives). It is not clear for me that this hypothesis can be fully ruled out. Indeed, the fact that the authors show a dependence on RecFOR for survival (Figure 3A) might indicate gap formation. It would be very helpful if this could be clarified, possibly by testing whether induction of nCas9 in stationary phase leads to a different dependence on the recombination enzymes: I would expect dependence on RecFOR and RecA but much less dependence on addAB (to clarify, I am not asking that the authors redo many experiments in stationary phase, but that they consider the possibility of gap repair and at least discuss it).

3)over-replication. In many panels showing MFA data (e. g.: 1D, 3B delta-RecF mutant, delta recR mutant, delta RecA mutant), there seems to be a higher signal toward oriC when a nick is present than when it is not produced (compare red and

blue curves to black ones). Yet over-replication is only mentioned for the delta-RecG mutant (which shows an increase only on a limited part of the chromosome). Could the authors comment on this phenomenon? Also, would it be to have the MFA panels in supplementary material in a larger format so that it is easier to assess subtle changes.

4) Effect of RecG: From the data in Figure 3A, RecG has an important role but is not fully essential. It also shows a smaller defect in replication by MFA. Also why is RecU not defective in replication but still very affected in viability? Maybe refer to the discussion here?

5) Role of RecFOR line 229-. References 50 and 51 do not seem to provide support for the role of recFOR after AddAB processing, as they deal with Gap repair in *E. coli* and the structure of RecFOR but not AddAB. I think <https://doi.org/10.1128/jb.01494-14> is a better reference, but it suggests that RecOR are necessary, not RecF (which seems to act only as a facilitator). The authors may wish to nuance their sentence (and refer to the discussion).

6) RecA loading: because the DNA is processed by AddAB, one would potentially expect RecA loading to be dependent on the position of Chi sites. Yet these positions to the nick and the locus test at -3kb in figure 4E and F are not mentioned. I am aware that there are many Chi sites on the *B. subtilis* chromosome, so this might not be relevant, but it should be mentioned somewhere. I am also surprised that the increase of RecA loading in the delta-RecG or delta-RecU mutant is not commented (Figure 4F).

7) Role of RecG: Figure S11 is hard to understand, and it is unclear how it connects to Figure 5 B. Please connect it to the relevant step in figure 3C (I assume post-synaptic) and to figure 5 B. also please add strand polarity.

8) Role of AddAB: line 325: the result of the double nuclease mutant of AddAB do not prove that end-resection does not happen. Resection in this mutant may be performed by another nuclease for example by RecJ. It would be interesting to check the viability of the triple mutant (double nuclease-dead+recJ). RecJ does not have helicase activity and would still require the helicase activity of AddAB, thus explaining the lethality in the helicase mutant. Regarding the MFA assay (Figure 5E): it would be useful to have a control with the WT on the same graph as the pattern of the double-nuclease mutant does seem affected (although less than the helicase mutant). Nevertheless, the result that the AddA-delta-nuc mutant can tolerate a delta-recG mutation is very interesting and strongly supports the model proposed by the authors.

9) Discussion; -line 355 and after: when describing the new model in the text, please use the very nice numbering of each step provided in figure 7 to facilitate the reading. -line 402-406: again, the statement should be nuanced as the role of RecJ or another nuclease in this particular background has not been ruled out. Line 442: The somewhat limited effect of the delta recG mutation on MFA should be commented on.

10) Methods:

-Strain construction: For strain construction of essential genes under a titrable promoter, can the exact strategy be explained briefly? I assume the titrable copy was inserted first in the WT background and then the endogenous copy removed? One or two sentences would be helpful.

-Why does the imaging medium contain glycerol and not glucose? This reduces replication and therefore the impact of the nicks.

-qPCR experiment: in figure 4B, SSB enrichment is stronger in delta-RecG than in WT. (and same with RecA figure 4F). Is it statistically significant? I don't think it is necessary to provide a statistical test for all the mutants and all the conditions, but it might be useful for a few mutants

Minor

-Figure 1E: please provide a scale bar. Is RecA-mNeogreen the only copy on the chromosome (I don't think so from the strain genotype?). If not, is this fusion functional? Why is RecA present only on the nucleoid and not the full cytoplasm as has been mostly reported in other papers?

-Cas9-ssrA and priA-ssrA: how are the fusions built exactly? Is there a linker between the protein and the tag? A very brief description would be helpful.

-Figure 2H: It is very nice that the weak nicking system supports replication. I am however puzzled by the lower DNA level just upstream of the nick. Could the author comment on that result?

-line 139. Consider adding "leading to replisome disassembly" after "replication for collapse" to clarify the expectation.

-Line 148. In the description of Fig. 2A, ChIP at OriC is mentioned, but I cannot find it in the panels B, C, D, E, F. I think the data on OriC is shown in Figure S4.

Line 160: I find the sentence commenting on the helicase behavior after a nick a bit hard to follow.

-line 205: The sentence is incorrect as mentioned above. RecA implication does not prove the formation of a DSB.

-line 214-218: The sequence could be more precise: RuvA and RuvB seem to have an effect at least for nicks on the leading strand. Moreover, it is not necessary surprising that RecJ and RecQ do not have an effect as they are usually only important when AddAB is not present.

-line 237 typo: seDSB

-line 355 and after: when describing the new model in the text, please use the very nice numbering of each step provided in figure 7 to facilitate the reading.

-line 378: the authors suggest that helicase slides into the ds template in an inactive conformation. Maybe add again a reference to reference 37?

-Figure 3 C, please add numbers to each steps (as in figure 7).

-Figure 5B and 6D: it is not easy to understand which recombination step these figures correspond to. I think they

correspond to the last step of figure 3C (post-synapsis), but it would be very helpful to clarify. Also in this figure and others (e. g. 5B), please make sure strand polarities (5', 3') are shown to facilitate understanding of the reasoning.

Meriem EL Karoui

(Remarks on code availability)

Reviewer #2

(Remarks to the Author)

Replication fork collapse scenario, when a progressing fork runs into a nick in template DNA and disintegrates, has been proposed back in 1994 to explain the highly-skewed distribution of Chi-sites in the bacterial chromosome, as well as synthetic lethality of mutations leading to increased DNA nicking with the defects in recombinational repair of double-strand DNA breaks (PMID: 8026461, PMID: 7565099, PMID: 10585965). The model has been tested in vivo in phage lambda chromosome, with site-specific nicking (your ref. #11), as well as in the *E. coli* chromosome, with non-specific nicking, by detecting oriC-containing chromosome fragments, but no terminus-containing ones (PMID: 16297932, PMID: 36243965). But it had not been tested yet with site-specific nicks in the bacterial chromosome using genome profiling.

The current manuscript from the Murray lab offers such test using the nicking Cas9 nuclease, in *B. subtilis* chromosome. The authors' thoughtful choice of the strain without prophages avoids complications associated with their SOS induction. The thorough and systematic study is based on three main readouts: 1) viability, 2) NGS chromosome profiling (= MFA); 3) ChIP-qPCR of nick-proximal DNA regions for replication and repair proteins. The readouts are orthogonal to each other, making conclusion solid. The paper is well-written and easy to read, even though it operates with a massive collection of data.

The main results, by Figures:

1. Induction of Cas9 defective for DNA cleavage has no phenotypes and is used as a negative control in all their assays. Induction of continuous Cas9 nicking reduces the plating titer by four orders of magnitude, no matter whether the nick is in the left or the right replicohores and no matter if it is in the leading strand or the lagging strand templates. The chromosome copy number profiles show no replication downstream of the nick (flat profile), with much lower copy number of the nick position (which the authors fail to comment on). Even though the nicking induces strong RecA filamentation (foci, bundles), the repair is futile, as the cells with constitutive nicking eventually lose their nucleoids. The authors interpret this all in terms of replication fork collapse to form double-strand ends — although they do not detect these one-ended breaks directly.

2A-G. By ChIP-PCR, they detect a huge over-presence of the replicative helicase, its three loaders, and PriA — all upstream of the nick. This is consistent with both fork stalling before the nicks or with them working on fork restart after the repair. Interestingly, while after the lagging strand template nicking the replicative helicase shows a very similar distribution, the leading strand nicking has even more helicase rolling into the area downstream of the nick. This is actually expected, as the replicative helicases are known to continue sliding into the contiguous duplex region.

2H-K. Since the continuous expression of the nickase kills the cells, for subsequent genetic studies of repair of the collapse forks, they developed a weak nicking system that causes only a mild trough in the WT cells. But even this weak nicking causes little replication downstream of the nick in the priA mutant and the viability decrease of 1,000.

3. Genetics of recombinational repair of double-strand breaks in *Bs* includes ds-end processing (addAB), RecA loading (recFOR), homology-guided strand invasion (recA) and the junction resolution (recG and recU). Using the weak nicking system, they show by MFA that all these functions are needed to various extent for the fork restoration. Interestingly, the effects range from severe (recAFOR) to intermediate (addAB) to modest (recGU). The ruvAB mutants (Fig. S8D) also show a modest effect similar to recU. The authors interpret this results with the classic scheme of restoration of the fork framework.

4AB. SSB is the first protein of the cell that binds ssDNA and therefore can be used as a ssDNA sensor. Although a replicative protein, it is also expected to participate in recombinational repair, handing over the DNA in need of repair to the recombinational repair machinery. Interestingly, although expected, SSB is not detected at the break site or even 1 kb away from the break. Instead, SSB distribution is maximal 2-4 kb away from the break and is still detectable 10 kb away. The nuclease that destroys the break-proximal ssDNA must be AddAB; interestingly, at position 3 kb away from the break, there is no SSB signal in the addAB mutant. Other rec mutants show either no change from WT in the recU mutant, 2-fold increase in the recFOR mutant (probably recA, too) and 3-fold increase in the recG mutant. Thus, AddAB not only degrades the break-proximal 1 kb, but also generates ssDNA in the break-distal 2-4 kb range.

4CDE. SSB protein has an interesting structure: the N-terminal 3/5th form a globular tetramer that ssDNA wraps around, while the C-terminal 2/5th is represented by an unstructured tail that ends with a charged tip (the C-terminal tip or CTT). This 6aa-tip can be deleted, and *Bacillus* cells are still alive, although grow slowly (*E. coli* would be dead). Remarkably, this Δ CTT ssb mutant is as deficient in repair of collapsed replication forks as the recA and recF mutants. Moreover, the Δ CTT mutant shows no defect in binding to ssDNA in vivo, but it is deficient in recruitment of RecF or RecA proteins to the collapsed fork.

4F. With the WT SSB, loading of RecA on the ssDNA upstream of the break site depends both on AddAB (to generate this DNA), as well as on the RecFOR, to replace SSB with RecA on this ssDNA.

5AB. The recG mutant, while showing a modest defect in repair of collapsed forks, also shows some overreplication in the 600 kb region upstream of the nick. The authors try to rationalize it in terms of replicative helicase mis-loading by PriA.

5CDEF. As its name suggests, the AddAB helicase/nuclease has the two activities. The nuclease activity is itself complex, with AddA-nuclease degrading the 3'-ending strand, while AddB-nuclease degrading the 5'-ending strand. The authors test mutants in individual activities and show that the helicase is important for collapsed fork repair (including SSB loading upstream of the nick), while the two nuclease activities are not. Interestingly, in the double-nuclease mutant, SSB is still loaded upstream of the nick, suggesting that DNA degradation in the cell is performed by ss-exo enzymes, like in *E. coli*.

6. Interestingly, the inactivation of AddA nuclease suppresses (weakly) the recG defect in collapsed fork repair, but the inactivation of AddB nuclease fails to do the same. The authors attach significance to this preliminary result.

Major points

Page 2, the first sentence of the Abstract and page 3, the first paragraph of the introduction — these need to be re-written to remove any mentioning of antibiotics. Oxidative DNA damage is indeed a real problem for all aerobic organisms and it does generate predominantly nicks (PMID: 12031895, PMID: 14637246). However, most antibiotics do not kill cells via oxidative damage — if they did, this would have been discovered long time ago, as a high sensitivity of recA mutants to ampicillin, kanamycin, chloramphenicol, rifampicin, etc. In fact, the recA mutants are only sensitive to DNA-targeting drugs, like Cipro or bleomycin, — so we should all stop perpetuating this nonsense linking all antibiotics to ROS production.

The deeply flawed studies of Kohansky and Collins (your ref. #3 and #5) have been long disqualified (PMID: 17761297, PMID: 22027063, PMID: 23836867, PMID: 23471410, PMID: 23471409, PMID: 24647480, PMID: 25666086, PMID: 28089288), and there is no need to propagate this popular misconception here. And there are a few antibiotics that do kill via oxidative DNA damage — for example, bleomycin and streptonigrin, — if the authors really feel the need to retain some antibiotics in the picture.

By the way, your ref#4 has nothing to do with antibiotics (unfortunately, it is also flawed).

In light of all this, the very last section of the Discussion (lines 477-485) should be also removed.

Page 14 and Fig. 6 — the suppression of the recG viability defect by inactivation of AddA nuclease, but not AddB nuclease, is an intriguing and a bit unexpected result, but it is so preliminary that it should not be part of this paper. First, this is the only serious result of the paper that is solely based on viability, with no follow up with the other two readouts (MFA, ChIP). Second, why is the double-nuclease mutant not tried/presented? Third, these addAB nuclease mutants should be combined with other rec mutations (especially recU) — what if they show the same distinct effects? I understand that this starts sounding like another paper — but this is exactly my point — do not try to fit this preliminary result into this otherwise solid and systematic study.

IMO, the recG result in Fig. 5A should still stay in this paper and can be possibly “finished” with a priA300-like suppression (no upstream overreplication is expected, judged by the *E. coli* results). The schemes in Fig. 5B and Fig. 6D are confusing because they are not linked to fork repair — and should be removed. The corresponding discussion section (lines 437-457) should also go, as there is really nothing to discuss at this point — the result is just a teaser.

Fig. 7 needs a (minor, yet important) modification. Due to the abundance of ss-exo enzymes in bacteria (PMID: 26442508), there is no such thing as ssDNA with unprotected ends there (the end polarity does not matter). Moreover, SSB binding, instead of protection, enhances ssDNA degradation by these processive enzymes (PMID: 16488881, PMID: 21572106). Therefore, stage III (SSB loading) cannot be correct. First, how come that SSB is loaded on the 3'-ending tail, but not on the 5'-ending tail? Second, both ss-tails will be SSB-complexed and immediately degraded by ss-exos, because the ends are still accessible.

Actually, the authors got it (partially) right in Fig. 6A — it is the looping of the 3'-ending strand (via controlling the end) by AddA that prevents it from being degraded by ss-exos (the incorrect aspect is preservation of the unprotected strand). Therefore, I suggest this important AddA-looping to be introduced at stage II (end processing) and kept until stage V (RecA recruitment). I am not sure that, in *E. coli*, RecBCD release from DNA by RecA polymerization was ever demonstrated — I encourage the authors to contact Steve Kowalczykowski with this question.

Thus, please remove all these unprotected ss-tails from stages III on (up to stage IX). Bacterial DNA metabolism simply does not tolerate them. Multiple ss-exo mutants are sick — the reason, although clear, is still unexplored experimentally.

Page 19, line 459 — the lack of requirements for Holliday junction migration/resolution for restart of collapsed forks sounds paradoxical, — but in this work this is likely due to the discrepancy between what is measured by the two readouts in question. The colony formation after overnight plating with the inducer (Fig. 3A) tells us that HJ resolution is important over multiple generations and breaks, while the replication restart after one or two double-strand end repair events (Fig. 3B) shows that HJ resolution is not THAT important. In fact, the fork restart per se probably does not even require HJ resolution — as reflected in the classic models of the process (for example, Fig. 23 of PMID: 10585965). However, long-term functionality of the chromosome does — and this would be likely seen if the time of “mild” Cas9 induction was increased from one hour to several hours. This should be discussed in the following section.

Minor points

Page 4, line 85 (also page 9, line 203, 206, 209 and so on, throughout the paper) — change “homologous recombination” to “recombinational repair”.

Pages 7-8, lines 165-173 — (this is FYI, not a request for more experiments) to actually distinguish between the fork stalling and fork restart, the same measurements for the helicase loaders should be done in the *priA* mutant, where the stalling could still happen, while reloading is blocked.

Page 9, lines 204-205 — this statement has to be modified to tone it down: cytology of the nucleoid cannot indicate specific DNA events — only that something dramatically bad has happened with the chromosome. Physical DNA studies could — in this case, PFGE could detect site-specific and one-sided double-strand breaks at the nicks.

Page 9, line 207 — there is nothing wrong with your ref#39 (published in 2000), other than that it is just a synopsis for the general public of the 1996 monograph “Recombinational Repair of DNA Damage” by Kuzminov (ISBN: 0-412-10671-X), or its 1999 bacteria-only version (PMID: 10585965). It is always better to cite the originals, — especially in this case, because the very phenomenon of replication fork instability (various mechanisms of disintegration) was recognized by Kuzminov (PMID: 7661854, PMID: 11459990), and the term “fork collapse” was also his contribution to the field (PMID: 8026461, PMID: 7565099).

Page 9, line 218 and Fig. S8D — The *ruvAB* mutants have a modest effect (reduced titer), similar to *recU*, so I would not dismiss them. You could tell it by their thinner-looking colonies.

Page 10, lines 225-226, “the hypothesis that homologous recombination is necessary to repair the DSB” — this WAS the hypothesis in the 1960s and through 1970s, but was already an accepted concept in the 1980s (remember the famous model of DSB repair PMID: 6380756?). Say instead: “These observations are consistent with recombinational repair of disintegrated replication forks” (PMID: 10585965).

Page 10, lines 229-231 — References #50 and 51 are not relevant here. In *E. coli*, RecFOR proteins are dispensable for DSB repair, because RecA loading onto ssDNA with concomitant SSB displacement is done by the RecBCD enzyme itself (PMID: 10731409, PMID: 16483938, PMID: 38261972). Better cite papers showing that RecFOR functions are required for DSB repair in *Bacillus* (for example, PMID: 15186413).

Page 10, line 233 — ref#53 is irrelevant to the statement it is supposed to support.

Page 10, line 237 — what is *seDSB*?

Page 14, line 338 — change “well” to “(barely)” (in parenthesis). Compared with WT, the colonies are barely growing.

Page 15, line 364 — delete “this type of”. Is there any other type of collapsed forks? There are several types of fork disintegration events (for example, Fig. 6A of PMID: 24806348); fork collapse is the only one that happens at nicks.

Page 16, lines 393-396 — to see the effect of *ss-exo* deficiencies, one needs to inactivate several major ones in a single mutant, — inactivation of any one of them has either no phenotype or a weak one (at least in *E. coli*). Therefore, as explained above in major points, it is safe to assume that in bacteria, any unprotected *ss-tail* is immediately degraded, resulting in blunt ends, no matter which template strand is nicked.

Page 17, lines 404-406 — I would modify the key role as to generate an end-protected *ss-loop* on the 3-ended strand (again, see above).

Page 17, line 407 — not only [SSB] is high near the fork, but the situation is likely the other way around: SSB forms a platform, on which the fork (or other processes, like *AddAB*-catalyzed unwinding) can function (see below).

Page 17, line 415 — yes, SSB is a protein hub, and this means it must form a platform for other proteins, as Lohman has long suggested. But since the bacterial nucleoid is a phase, this also means that the SSB platforms float on the surface of this phase and, besides handling ssDNA, serve as the nucleation points for assembly of secondary platforms, like the *RecA* one (PMID: 38358278). Fittingly, your *SSB Δ6* mutation that should disrupt the formation of such a platform, has terrible effects on the subsequent recombinational repair (although it does not affect ssDNA binding).

Page 18, lines 430-433 — the *RecF-DnaN* interactions, if confirmed in *Bacillus*, will be relevant only for the daughter-strand gap repair. Here the authors are trying to apply these interactions to the double-strand end repair, but there is no replisome around — it was out of the picture the moment replication fork has collapsed. Therefore, I suggest the authors just point out that at the moment it is not clear what invites *RecF* to the *AddAB*-held ssDNA loop. I would bet on direct interactions between *AddAB* and *RecF*.

Page 19, lines 455-457 — please clean up this word salad. Collapsed forks are not “resolved” (you do not have to repeat other people’s mis-statements), they need to be repaired and restarted (in prokaryotes) or just repaired (= the fork framework reassembled) without the restart (in eukaryotes, where no restart machinery exists). Further, replisome cannot collapse, only the fork can — and then the replisome had to lose contact with the resulting DNA pieces (and is likely disassembled).

Page 32, line 753 — the helicase is shown as a circle, rather than hexagon. Also, in the panel G itself, replace “Helicase collapses” (what is this?) with “Helicase runs off”.

All chromosome profiles with active nicking — how would the authors explain that the profile minima coincide with the position of the nick? Is it a real degradation of the template DNA around the nick? Or is it incomplete isolation of DNA for MFA, due to the region being complexed with recombinational repair proteins/enzymes?

Fig. 3B — for comparison purposes, I would repeat there the panels 2H (WT) and 2K (priA). Besides, there is space there for two more panels.

Fig. 5B scheme is confusing, rather than clarifying. For example, do these events happen after fork collapse and repair? Or independent of it, at an intact replication fork? Fig. S11 does not help either (besides, the two structures should be the same, right?).

Fig. 8C — “addO” is likely “recO”

(Remarks on code availability)

Reviewer #3

(Remarks to the Author)

This is a very interesting and important set of experiments that define some of the events that occur when a bacterial replication fork hits a template discontinuity. The work is innovative, provides new information, and greatly solidifies speculation based on other, less direct observations. In general, the work provides a broader and much clearer view of a very important bacterial process. The proteins defined as essential for double strand break repair are not surprising but some of the proteins that are NOT required may surprise. The efforts to define RecG participation break some significant new ground and the observation of a RecO-SSB interaction requirement is nicely done. The absence of a requirement for the AddAB nuclease activities is also an important contribution.

The system the authors have developed is quite complex and more effort is needed to lay out all the moving parts clearly. Most of my comments are to that end.

Fig 1C When the nickase is induced in normal cells, viability declines by some 5 orders of magnitude. But a few cells survive. In those cells, has the nickase been inactivated or the target site mutated?

Fig 2A. Some distance indication is needed. How far from the nick site are these various positions? There are many references to upstream and downstream of the nick. How far upstream and downstream in the exp of Fig 2?

Fig S6B. Is xylose present here?

In coli, RecBCD is able to load RecA on its own and the requirement here for RecFOR may surprise some readers. If AddAB does not load RecA on its own, a discussion of this difference would be useful. Also, a mention of RecU function when it first appears would be useful.

(Remarks on code availability)

The referenced structures are readily available and easily accessed.

Version 1:

Reviewer comments:

Reviewer #1

(Remarks to the Author)

I would like to thank the authors for carefully addressing all the comments and providing additional data that strengthen the manuscript. Without listing all the changes, I was particularly impressed by the experiment involving the RexAB protein of *S. aureus*. The observation that the SSB recruitment position is altered, aligning with the position of the *S. aureus* Chi sites, strongly supports the authors' conclusions. Furthermore, the rewriting has greatly clarified the text.

I congratulate the authors on an excellent paper reporting very exciting results!

My only minor suggestion is to use the term “the replicative helicase” in the abstract (line 25) instead of “helicase” to avoid confusion.

Meriem El Karoui

(Remarks on code availability)

Reviewer #2

(Remarks to the Author)

Good job, authors! I really enjoyed this paper.

Minor points

Page 12, lines 281-283: "... or that AddAB helicase activity is necessary and sufficient to separate the two strands of the seDSB and facilitate SSB loading onto ssDNA at a 3'-end." — This is unlikely, as the 5'-ending strand still needs to be removed — otherwise it will poison the subsequent RecA-mediated homology search by providing the proximal "homology". It will be preferentially removed by (unspecified) ssDNA exonucleases, as long as the 3'-ending strand is secured in a loop by AddAB.

Page 12, lines 286/287: "The requirement of the RecFOR system for recombinational repair (Fig. 2) implied that SSB is bound to the strand terminating with a 3'-end." — Why the 3'-end? I am not sure where is this specific polarity coming from. Hypothetically, it could be through the AddAB-RecFOR physical interactions, but there is no evidence about it yet.

Page 20, lines 490-494 — these vague explanations of the Δ recU mutant defects in chromosome segregation are completely unnecessary. As I pointed out in the previous round of critique, restart of collapsed replication forks after reassembly by recombinational repair was always proposed to be independent of the Holliday junction resolution behind the fork (for example, Fig. 23 of PMID: 10585965), but Holliday junction resolution is required to finish the repair — otherwise the two sister chromosomes will remain linked together and unsegregatable! And the XerCD monomerization system will not help in this case.

In fact, the authors should argue in Discussion that this result supports the idea that fork restart is independent of HJ resolution, but HJ resolution is required for completion of repair via disconnecting the sister chromatids — otherwise chromosome partitioning problems.

Miscellaneous

Page 16, line 375 — change "genome" to "template DNA". "Genome" refers to information content of DNA, rather than to DNA structure.

Page 16, line 381 "intolerant" — is "tolerant" meant there?

Page 18, line 449 "replication repair" — was "recombinational repair" meant here?

Andrei Kuzminov

(Remarks on code availability)

Reviewer #3

(Remarks to the Author)

The response to referees is comprehensive and a strong study has been rendered stronger as a result. However, one error has been added in the revision that should be removed. On line 246, the authors have now mentioned a RecFOR complex. There is no such thing and its mention can be misleading. As no RecFOR complex has ever been described (there are RecOR or RecFR complexes but no RecFOR), the word "complex" here needs to be changed to "system".

(Remarks on code availability)

Response to reviewers: NCOMMS-25-28666-T

The authors would like to sincerely thank the Reviewers for their constructive questions, comments, and criticisms. We would like to highlight two major changes to the revised version of the manuscript. First, following the recommendation of Reviewer #2, we have removed the genetic analysis and discussion of RecG acting specifically on a subclass of replication forks generated by AddAB mutants. Second, we have generated new strains showing that RecG and RecU are both enriched upstream of a nick, supporting their direct role in recombinational repair.

Reviewer #1 (Remarks to the Author):

In the manuscript entitled "Rescuing bacterial genome replication: essential functions to repair a double-strand break and restart DNA synthesis", Winterhalter and colleagues use a clever genetic system based on a modified Cas9 enzyme to induce DNA nicks in a site-specific manner. Such a nick mimics the state of DNA after cleavage of a damaged base (i.e. 8-oxoguanine). It therefore opens up the possibility of performing a mechanistic analysis of the repair pathways involved, particularly concerning how such a nick is repaired when converted into a DNA double-strand break when a replication fork arrives.

The authors utilise the ability to modify the cleavage strand of Cas9 to produce a nick on either the leading or lagging strand, an elegant strategy that further enables them to delve into the details of the repair mechanisms. They use multiple genetic strategies (deletion and separation of function mutants, degron tags, inducible complementation of essential genes, etc) combined with genome-wide Marker Frequency Analysis and Chip qPCR to provide a very comprehensive picture of the pathways involved. As far as I can tell, this is the first time that site-specific nicks have been analysed in live bacteria, and the work presented here is novel and important. The manuscript is well written and the figures clear and well presented. There are however some questions that should be clarified and are described below.

Major comments:

1)The introduction focuses on the base damage caused by ROS due to antibiotics. However, there are many other possibilities to create damage (indeed, the graphical abstract mentioned the immune system) and these should be mentioned too. Moreover, a brief mention of Base excision repair (which might be expected to be involved in the repair of the nick produced by Cas9?) would be helpful.

The Introduction has been rewritten to focus on a broader range of DNA repair systems that share nucleotide excision as an intermediate step, thereby generating a single-strand discontinuity in the DNA backbone (lines 42-51). The graphical abstract has been amended to better reflect the revised narrative of the study.

2)Another important point is to clarify whether the authors expect that the only type of DNA damage caused by their system is a replication-dependent double-strand break (DSB) or whether other types of damage can occur? In the first paragraph of the results section, the author provide compelling evidence through MFA that replication is impaired by the nicks and this will definitely result in DSBs. However, the evidence provided does not prove that the only type of damage is a DSB. Localisation of RecA and formation of spots and bundles (from line 124) does not by itself prove that the damage is a DSB because RecA catalyses homologous recombination from DSB but also from single-strand gaps (RecFOR pathway). As far as I understand, the system creates a nick and not a single-nucleotide gap (I.e. the DNA

backbone is cut but the nucleotides are still all present), but this is not clearly stated. Moreover, it is conceivable that the nick could lead to degradation of a few nucleotides on the nicked strand that would create a gap. Similarly, as shown by the authors on the diagram of figure 1A (right panel), even after replication, the strand that does not have a DSB shows a gap. Has the possibility of gap formation been completely ruled out?

In this case, one could conceivably observe repair by homologous recombination through the RecFOR gap repair pathway independently of replication (for example one can conceive that a gap is repaired before the replication fork arrives). It is not clear for me that this hypothesis can be fully ruled out. Indeed, the fact that the authors show a dependence on RecFOR for survival (Figure 3A) might indicate gap formation. It would be very helpful if this could be clarified, possibly by testing whether induction of nCas9 in stationary phase leads to a different dependence on the recombination enzymes: I would expect dependence on RecFOR and RecA but much less dependence on AddAB (to clarify, I am not asking that the authors redo many experiments in stationary phase, but that they consider the possibility of gap repair and at least discuss it).

These are great points. We agree that the size of the ssDNA gap generated following phosphodiester bond cleavage by nCas9 could be larger than a simple nick. We also agree that multiple types of DNA damage may be present at the site of nicking. Critically however, following nick induction, recruitment of both SSB (Fig. 5B) and RecA (Fig. 6F) were dependent upon AddAB (which requires a DNA end as a substrate for helicase and nuclease activity - PMID 20116346). Therefore, we believe the data favours a model where DSB formation is the dominant lesion and that RecFOR is essential for RecA loading at these sites.

To test the hypothesis that lethal DNA damage generated at a nick was replication-dependent, we performed a timecourse experiment where nCas9 was induced for 4 hours, either during logarithmic growth or during entry to stationary phase. Plating assays showed that nCas9 nicking was only lethal during exponential growth (new Fig. S1E), consistent with replication dependent DSB formation.

Within the text, as well as emphasizing the connection between AddAB and its substrate (a DNA end), we have rewritten the Discussion (lines 459-467): “Historically the RecFOR system has been proposed to mediate ssDNA gap-filling, rather than recombination at a DSB⁶¹. While it is possible that multiple types of DNA lesions are associated with nicks generated by nCas9 in *B. subtilis*, we note that recruitment of SSB and RecA was dependent on AddAB (Figs. 5B, 6F). Because AddAB-dependent end-processing is not associated with models for recombinational repair of ssDNA gaps, the data presented here is most consistent with RecFOR acting within the proposed seDSB repair pathway (Fig. 8). Supporting this viewpoint, AddAB has been shown to be epistatic to RecFOR in *Thermus thermophilus*¹⁰⁰ and RecFOR is essential for DSB repair in *Deinococcus radiodurans*¹⁰¹”.

3)over-replication. In many panels showing MFA data (e. g.: 1D, 3B delta-RecF mutant, delta recR mutant, delta RecA mutant), there seems to be a higher signal toward oriC when a nick is present than when it is not produced (compare red and blue curves to black ones). Yet over-replication is only mentioned for the delta-RecG mutant (which shows an increase only on a limited part of the chromosome). Could the authors comment on this phenomenon? Also, would it be to have the MFA panels in supplementary material in a larger format so that is it easier to assess subtle changes.

As suggested, MFA panels including enlarged regions of interest have been added to illustrate replication profile phenotypes of recombination mutants, either in the absence (Fig. S9) or in presence of nicking (Figs. S10, S11, lines 231-241).

4) Effect of RecG: From the data in Figure 3A, RecG has an important role but is not fully essential. It also shows a smaller defect in replication by MFA. Also why is RecU not defective in replication but still very affected in viability? Maybe refer to the discussion here?

While we cannot be certain why the RecG mutation is less defective, in the revised manuscript we have emphasised how RecA recruitment is RecG-independent (in contrast to AddAB and RecFOR, which all display the strongest growth defect following nicking - lines 348-350). We have also provided new data showing that RecG is enriched upstream of a nick (Fig. 7A), consistent with a direct role in recombinational repair.

We agree that the role of RecU in recombinational repair is unclear. We provide new data showing that the $\Delta recU$ mutant displays significant chromosome segregation abnormalities (Fig. 7B), raising the possibility that the growth defect following nCas9 nicking is a synthetic phenotype. However, we have also provided new data showing that RecU is enriched upstream of a nick (Fig. 7C), consistent with a direct role in recombinational repair. The Results (lines 360-372) and Discussion (lines 487-497) have been amended to reflect these findings.

5) Role of RecFOR line 229-. References 50 and 51 do not seem to provide support for the role of recFOR after AddAB processing, as they deal with Gap repair in *E. coli* and the structure of RecFOR but not AddAB. I think DOI 10.1128/jb.01494-14 is a better reference, but it suggests that RecOR are necessary, not RecF (which seems to act only as a facilitator). The authors may wish to nuance their sentence (and refer to the discussion).

We have added the recommended reference and rewritten the section about RecFOR in Discussion (lines 459-467). We have also amended the text to be more specific (lines 245-247): "The single-stranded DNA (ssDNA) is recognised and bound by the single-stranded binding protein (SSB), which must be removed by the RecFOR complex to allow RecA filament formation (Fig. 3C(IV-V))".

6) RecA loading: because the DNA is processed by AddAB, one would potentially expect RecA loading to be dependent on the position of Chi sites. Yet these positions to the nick and the locus test at -3kb in figure 4E and F are not mentioned. I am aware that there are many Chi sites on the *B. subtilis* chromosome, so this might not be relevant, but it should be mentioned somewhere. I am also surprised that the increase of RecA loading in the delta-RecG or delta-RecU mutant is not commented (Figure 4F).

The reviewer raised an important question about the presence of *chi* sites. We have added the locations of these sites to Fig. 5A where we report the enrichment of SSB. Moreover, to investigate the model of *chi* sites in recombinational repair, we functionally replaced AddAB with the homologous protein complex from *Staphylococcus aureus* (RexAB - Fig. S12E), which recognises a distinct *chi* sequence. Analysis of SSB enrichment in this strain showed that the binding profile is shifted further upstream to a position proximal to the first *S. aureus chi* site (Fig. 5A), consistent with a functional link between pausing at *chi* sites and producing ssDNA for recombination.

We have modified the description of His-RecA in the results to address this point (lines 348-350): "ChIP enrichment of RecA was independent of RecG (Fig. 6F), suggesting that RecG

acts downstream of RecA-mediated strand exchange, consistent with proposed activities in DNA remodelling”.

7) Role of RecG: Figure S11 is hard to understand, and it is unclear how it connects to Figure 5 B. Please connect it to the relevant step in figure 3C (I assume post-synaptic) and to figure 5 B. also please add strand polarity.

As described at the start of our Response to Reviewers, this supplementary figure and corresponding sections have been removed from the revised manuscript to refocus the narrative on post-synapsis requirements of RecG and RecU for replication restart.

8) Role of AddAB: line 325: the result of the double nuclease mutant of AddAB do not prove that end-resection does not happen. Resection in this mutant may be performed by another nuclease for example by RecJ. It would be interesting to check the viability of the triple mutant (double nuclease-dead+recJ). RecJ does not have helicase activity and would still require the helicase activity of AddAB, thus explaining the lethality in the helicase mutant. Regarding the MFA assay (Figure 5E): it would be useful to have a control with the WT on the same graph as the pattern of the double-nuclease mutant does seem affected (although less than the helicase mutant). Nevertheless, the result that the AddA-delta-nuc mutant can tolerate a delta-recG mutation is very interesting and strongly supports the model proposed by the authors.

As suggested we have tested viability of AddAB nuclease variants combined with a *recJ* deletion. Spot titre analysis showed that there was no decrease in CFU in the triple nuclease mutant compared to controls (Fig. S12D). These data further strengthen the model proposed and we adapted wording throughout the manuscript supporting the statement (lines 277-280): “Deletion of *recJ* in conjunction with either *addA*^{ΔNUC}, *addB*^{ΔNUC}, or *addA*^{ΔNUC}*addB*^{ΔNUC} did not reduce the number of CFU following nicking (Fig. S12D), indicating that RecJ nuclease activity is not required under these conditions”.

As recommended, we have also added a wild-type control to MFA profiles of AddAB variants (Figs. 4D, S12A-B).

9) Discussion; -line 355 and after: when describing the new model in the text, please use the very nice numbering of each step provided in figure 7 to facilitate the reading.

Done as suggested.

-line 402-406: again, the statement should be nuanced as the role of RecJ or another nuclease in this particular background has not been ruled out.

Please refer to response to comment #8 above regarding the triple AddAB/RecJ nuclease mutants (Fig. S12D). Moreover, in the revised manuscript we have attempted to emphasise the critical role of DNA unwinding by a helicase, rather than drawing conclusions regarding DNA degradation (lines 445-447): “Together these results raise the unexpected possibility that during recombinational repair at a nick, the key activity during end-processing is to separate the strands and create a ssDNA substrate for SSB (Figs. 5C, 8(II-III))”.

Line 442: The somewhat limited effect of the delta recG mutation on MFA should be commented on.

Similar to our response to comment #4 above, while we cannot be certain why the RecG mutation is less defective, we suspect it relates to a late temporal role in recombination (e.g. post D-loop formation). We have also provided new data showing that RecG is enriched upstream of a nick (Fig. 7A), consistent with a direct role in recombinational repair.

10) Methods: -Strain construction: For strain construction of essential genes under a titrable promoter, can the exact strategy be explained briefly? I assume the titrable copy was inserted first in the WT background and then the endogenous copy removed? One or two sentences would be helpful.

We have added technical details to the Methods (lines 608-616).

-Why does the imaging medium contain glycerol and not glucose? This reduces replication and therefore the impact of the nicks.

In *Bacillus subtilis*, growth and DNA replication initiation frequency are nearly identical in media containing either sugar as a carbon source. Moreover, in the experiments reported here, defined synthetic media contain casamino acids supporting multifork replication (PMID 25340815).

-qPCR experiment: in figure 4B, SSB enrichment is stronger in delta-RecG than in WT. (and same with RecA figure 4F). Is it statistically significant? I don't think it is necessary to provide a statistical test for all the mutants and all the conditions, but it might be useful for a few mutants

As suggested, we performed statistical analyses of the results and detected a significant SSB enrichment increase for $\Delta recG$ strains and a $\Delta recF$ variant. The text has been modified accordingly (line 842-843): "note a significant increase in SSB enrichment in some of mutants tested under these conditions".

Minor

-Figure 1E: please provide a scale bar. Is RecA-mNeongreen the only copy on the chromosome (I don't think so from the strain genotype?). If not, is this fusion functional? Why is RecA present only on the nucleoid and not the full cytoplasm as has been mostly reported in other papers?

Scale bars have been provided to all microscopy images.

Further description of the *mNG-recA* cassette is given in the Results (lines 126-129): "To determine whether DNA damage was generated following nCas9 induction, localisation of an ectopic mNeonGreen-RecA reporter (mNG-RecA) was observed in live cells using fluorescence microscopy (endogenous *recA* is present in these merodiploid strains)" and in the Methods (lines 619-620): "the fluorescent mNG-RecA DNA break reporter was constructed by N-terminal fusion via a short serine/glycine linker (SG₃)".

The localisation of mNG-RecA shown in this manuscript is consistent with the literature in *B. subtilis* (PMID 17229847, fluorescent RecA associates with the nucleoid in 'unstressed' conditions). Prior fluorescent fusions to RecA used distinct fluorophores, linkers, and expression systems, which may also explain variability across fluorescently tagged RecA constructs. Here, mNG-RecA specifically assembles into foci or bundles upon introduction of

nicks, which indicates that the *mNG-recA* fusion is functional as a DNA break reporter (Fig. 1E, S3B).

-Cas9-*ssrA* and *priA-ssrA*: how are the fusions built exactly? Is there a linker between the protein and the tag? A very brief description would be helpful.

This has been addressed above (lines 617-620): “For the construction of protein fusions, *ssrA* tags (AANDENYSENYALAA) were directly fused to C-terminal ends of target proteins (e.g. Cas9-*ssrA* variants and *PriA-ssrA*) [...]”.

-Figure 2H: It is very nice that the weak nicking system supports replication. I am however puzzled by the lower DNA level just upstream of the nick. Could the author comment on that result?

We suspect that this drop reflects wild-type processing of the *seDSB* in a mixed population of cells at different stages of recombinational repair and replication restart. The results are now better described (lines 188-191): “MFA showed that DNA synthesis continued after a one-hour xylose induction, consistent with low tolerable levels of nickase activity (Fig. 2H, note a reduction in read depth upstream of nickase target sites, likely reflecting recombinational repair)”.

-line 139. Consider adding “leading to replisome disassembly” after “replication for collapse” to clarify the expectation.

This statement was changed to (lines 136-138): “These results are consistent with the model that the bacterial replisome generates DNA damage upon encountering of a nick in either strand of the parental DNA duplex”.

-Line 148. In the description of Fig. 2A, ChIP at *OriC* is mentioned, but I cannot find it in the panels B, C, D, E, F. I think the data on *OriC* is shown in Figure S4.

For clarity, we have removed references to enrichment at *oriC* in Fig. 2A and clarified (lines 151-153): “Relative protein enrichment proximal to the nick was normalised to a control locus located on the opposite chromosome arm equidistant from *oriC* (+90, Fig. 2A)”.

Line 160: I find the sentence commenting on the helicase behavior after a nick a bit hard to follow.

This paragraph has been rewritten to aid clarity (lines 157-167).

-line 205: The sentence is incorrect as mentioned above. *RecA* implication does not prove the formation of a DSB.

This is related to comment #2 above. The wording has been changed (lines 211-212): “Genetic and cytological data suggest that when a replisome encounters a nick, it generates a substrate with ssDNA that is sensed by *RecA* (Fig. 1C-E)”.

-line214-218: The sequence could be more precise : *RuvA* and *RuvB* seem to have an effect

at least for nicks on the leading strand. Moreover, it is not necessary surprising that RecJ and RecQ do not have an effect as they are usually only important when AddAB is not present.

While RuvAB mutants display a mild growth defect in the presence of nicks (Fig. S8D), the number of CFUs observed under these conditions is comparable to the parent control. The manuscript is now consistent with the phrase (lines 228-230): "Gene products involved in Holliday junction migration (RuvA, RuvB)⁴⁷ displayed a small colony phenotype but not a loss of CFUs (Fig. S8D), suggesting they play a notable yet less critical role in the repair pathway".

-line237 typo: seDSB

The term single-end double-strand break (seDSB) has been defined (lines 50-51) and the text has been changed throughout the manuscript.

-line 355 and after: when describing the new model in the text, please use the very nice numbering of each step provided in figure 7 to facilitate the reading.

Done.

-line 378: the authors suggest that helicase slides into the ds template in an inactive conformation. Maybe add again a reference to reference 37?

This reference has been added.

-Figure 3 C, please add numbers to each steps (as in figure 7).

Done.

-Figure 5B and 6D: it is not easy to understand which recombination step these figures correspond to. I think they correspond to the last step of figure 3C (post-synapsis), but it would be very helpful to clarify. Also in this figure and others (e. g. 5B), please make sure strand polarities (5', 3') are shown to facilitate understanding of the reasoning.

These figure panels were removed from the revised manuscript. The RecG sections of the manuscript (Results/Discussion) have been rewritten accordingly.

Reviewer #2 (Remarks to the Author):

Replication fork collapse scenario, when a progressing fork runs into a nick in template DNA and disintegrates, has been proposed back in 1994 to explain the highly-skewed distribution of Chi-sites in the bacterial chromosome, as well as synthetic lethality of mutations leading to increased DNA nicking with the defects in recombinational repair of double-strand DNA breaks (PMID: 8026461, PMID: 7565099, PMID: 10585965). The model has been tested in vivo in phage lambda chromosome, with site-specific nicking (your ref. #11), as well as in the E. coli chromosome, with non-specific nicking, by detecting oriC-containing chromosome fragments, but no terminus-containing ones (PMID: 16297932, PMID: 36243965). But it had not been tested yet with site-specific nicks in the bacterial chromosome using genome profiling. The current manuscript from the Murray lab offers such test using the nicking Cas9 nuclease, in B. subtilis chromosome. The authors' thoughtful choice of the strain without prophages avoids complications associated with their SOS induction. The thorough and systematic study is based on three main readouts: 1) viability, 2) NGS chromosome profiling (= MFA); 3) ChIP-qPCR of nick-proximal DNA regions for replication and repair proteins. The readouts are orthogonal to each other, making conclusion solid. The paper is well-written and easy to read, even though it operates with a massive collection of data.

The main results, by Figures:

1. Induction of Cas9 defective for DNA cleavage has no phenotypes and is used as a negative control in all their assays. Induction of continuous Cas9 nicking reduces the plating titer by four orders of magnitude, no matter whether the nick is in the left or the right replichores and no matter if it is in the leading strand or the lagging strand templates. The chromosome copy number profiles show no replication downstream of the nick (flat profile), with much lower copy number of the nick position (which the authors fail to comment on). Even though the nicking induces strong RecA filamentation (foci, bundles), the repair is futile, as the cells with constitutive nicking eventually lose their nucleoids. The authors interpret this all in terms of replication fork collapse to form double-strand ends — although they do not detect these one-ended breaks directly.

We have modified the text throughout to be more precise regarding the interpretation of the events and DNA structures at the nicking sites (e.g. broken replication forks and replisome inactivation). Note that following the discovery that AddAB helicase activity is necessary for recombinational repair, we explain why this result favours a model for a repair pathway involving a DSB and refer to a single-end DSB (seDSB) thereafter (lines 239-241).

2A-G. By ChIP-PCR, they detect a huge over-presence of the replicative helicase, its three loaders, and PriA — all upstream of the nick. This is consistent with both fork stalling before the nicks or with them working on fork restart after the repair. Interestingly, while after the lagging strand template nicking the replicative helicase shows a very similar distribution, the leading strand nicking has even more helicase rolling into the area downstream of the nick. This is actually expected, as the replicative helicases are known to continue sliding into the contiguous duplex region.

2H-K. Since the continuous expression of the nickase kills the cells, for subsequent genetic studies of repair of the collapse forks, they developed a weak nicking system that causes only a mild trough in the WT cells. But even this weak nicking causes little replication downstream of the nick in the priA mutant and the viability decrease of 1,000.

3. Genetics of recombinational repair of double-strand breaks in *Bs* includes ds-end processing (*addAB*), RecA loading (*recFOR*), homology-guided strand invasion (*recA*) and the junction resolution (*recG* and *recU*). Using the weak nicking system, they show by MFA that all these functions are needed to various extent for the fork restoration. Interestingly, the effects range from severe (*recAFOR*) to intermediate (*addAB*) to modest (*recGU*). The *ruvAB* mutants (Fig. S8D) also show a modest effect similar to *recU*. The authors interpret this results with the classic scheme of restoration of the fork framework.

Note that following Reviewer #1's suggestions, we have expanded the description of MFA results in this section (Figs. S9, S10, S11).

4AB. SSB is the first protein of the cell that binds ssDNA and therefore can be used as a ssDNA sensor. Although a replicative protein, it is also expected to participate in recombinational repair, handing over the DNA in need of repair to the recombinational repair machinery. Interestingly, although expected, SSB is not detected at the break site or even 1 kb away from the break. Instead, SSB distribution is maximal 2-4 kb away from the break and is still detectable 10 kb away. The nuclease that destroys the break-proximal ssDNA must be *AddAB*; interestingly, at position 3 kb away from the break, there is no SSB signal in the *addAB* mutant. Other *rec* mutants show either no change from WT in the *recU* mutant, 2-fold increase in the *recFOR* mutant (probably *recA*, too) and 3-fold increase in the *recG* mutant. Thus, *AddAB* not only degrades the break-proximal 1 kb, but also generates ssDNA in the break-distal 2-4 kb range.

Please note that we have provided new experimental evidence suggesting that a critical determinant of SSB recruitment is the end-processing machinery encountering a *chi* site (Figs. 5A, S12E). This is consistent with *in vitro* evidence demonstrating that *AddAB* undergoes a helicase molecular switch upon *chi* recognition (PMIDs 17570399 and 24670664) and further strengthens the model proposed for the role of end-processing in this pathway.

4CDE. SSB protein has an interesting structure: the N-terminal 3/5th form a globular tetramer that ssDNA wraps around, while the C-terminal 2/5th is represented by an unstructured tail that ends with a charged tip (the C-terminal tip or CTT). This 6aa-tip can be deleted, and *Bacillus* cells are still alive, although grow slowly (*E. coli* would be dead). Remarkably, this Δ CTT *ssb* mutant is as deficient in repair of collapsed replication forks as the *recA* and *recF* mutants. Moreover, the Δ CTT mutant shows no defect in binding to ssDNA *in vivo*, but it is deficient in recruitment of *RecF* or *RecA* proteins to the collapsed fork.

Please note that the immunoprecipitated protein here was His-RecO (not *RecF*), which is consistent with the SSB:RecO interaction reported *in vitro* (PMID 21170359).

4F. With the WT SSB, loading of *RecA* on the ssDNA upstream of the break site depends both on *AddAB* (to generate this DNA), as well as on the *RecFOR*, to replace SSB with *RecA* on this ssDNA.

5AB. The *recG* mutant, while showing a modest defect in repair of collapsed forks, also shows some overreplication in the 600 kb region upstream of the nick. The authors try to rationalize it in terms of replicative helicase mis-loading by *PriA*.

As suggested by the Reviewer (comment #2 below), this RecG panel has been removed for overall clarity and replaced by ChIP data showing that RecG is specifically enriched upstream of nick sites (Fig. 7A).

5CDEF. As its name suggests, the AddAB helicase/nuclease has the two activities. The nuclease activity is itself complex, with AddA-nuclease degrading the 3'-ending strand, while AddB-nuclease degrading the 5'-ending strand. The authors test mutants in individual activities and show that the helicase is important for collapsed fork repair (including SSB loading upstream of the nick), while the two nuclease activities are not. Interestingly, in the double-nuclease mutant, SSB is still loaded upstream of the nick, suggesting that DNA degradation in the cell is performed by ss-exo enzymes, like in E. coli.

We have added a section about the major exonuclease RecJ and the results support the phrase (lines 280-283): “Based on these observations, we hypothesise that either another nuclease(s) promotes resection of the seDSB to generate a 3'-tail, or that AddAB helicase activity is necessary and sufficient to separate the two strands of the seDSB and facilitate SSB loading onto ssDNA with 3'-end”.

6. Interestingly, the inactivation of AddA nuclease suppresses (weakly) the recG defect in collapsed fork repair, but the inactivation of AddB nuclease fails to do the same. The authors attach significance to this preliminary result.

As suggested by the Reviewer (comment #2 below), this subsection of AddAB with RecG has been removed for overall clarity and replaced by ChIP data showing that RecG is specifically enriched upstream of nick sites (Fig. 7A).

Major points

(1) Page 2, the first sentence of the Abstract and page 3, the first paragraph of the introduction — these need to be re-written to remove any mentioning of antibiotics. Oxidative DNA damage is indeed a real problem for all aerobic organisms and it does generate predominantly nicks (PMID: 12031895, PMID: 14637246). However, most antibiotics do not kill cells via oxidative damage — if they did, this would have been discovered long time ago, as a high sensitivity of recA mutants to ampicillin, kanamycin, chloramphenicol, rifampicin, etc. In fact, the recA mutants are only sensitive to DNA-targeting drugs, like Cipro or bleomycin, — so we should all stop perpetuating this nonsense linking all antibiotics to ROS production. The deeply flawed studies of Kohansky and Collins (your ref. #3 and #5) have been long disqualified (PMID: 17761297, PMID: 22027063, PMID: 23836867, PMID: 23471410, PMID: 23471409, PMID: 24647480, PMID: 25666086, PMID: 28089288), and there is no need to propagate this popular misconception here. And there are a few antibiotics that do kill via oxidative DNA damage — for example, bleomycin and streptonigrin, — if the authors really feel the need to retain some antibiotics in the picture. By the way, your ref#4 has nothing to do with antibiotics (unfortunately, it is also flawed). In light of all this, the very last section of the Discussion (lines 477-485) should be also removed.

As suggested, the Introduction has been refocused to consider single-strand discontinuities arising from DNA repair events (related to Reviewer #1 comment #1).

(2) Page 14 and Fig. 6 — the suppression of the recG viability defect by inactivation of AddA

nuclease, but not AddB nuclease, is an intriguing and a bit unexpected result, but it is so preliminary that it should not be part of this paper. First, this is the only serious result of the paper that is solely based on viability, with no follow up with the other two readouts (MFA, ChIP). Second, why is the double-nuclease mutant not tried/presented? Third, these addAB nuclease mutants should be combined with other rec mutations (especially recU) — what if they show the same distinct effects? I understand that this starts sounding like another paper — but this is exactly my point — do not try to fit this preliminary result into this otherwise solid and systematic study. IMO, the recG result in Fig. 5A should still stay in this paper and can be possibly “finished” with a priA300-like suppression (no upstream overreplication is expected, judged by the *E. coli* results). The schemes in Fig. 5B and Fig. 6D are confusing because they are not linked to fork repair — and should be removed. The corresponding discussion section (lines 437-457) should also go, as there is really nothing to discuss at this point — the result is just a teaser.

We appreciate the reviewer’s constructive feedback. As suggested, we have removed these data from the manuscript.

The suggestion to employ a PriA300-like mutant to test RecG essentiality and replication restart is insightful. Unfortunately however, we and others have found that this ATPase mutant is not viable in *B. subtilis* (PMID 11679082). Instead, we have provided further details of MFA features associated with the Δ recG strain background (Figs. S9C-D, S11A) and performed ChIP showing that FLAG-RecG is recruited upstream of nicks (Fig. 7A). Together, these changes build on previously reported results and strengthen the narrative of the manuscript.

(3) Fig. 7 needs a (minor, yet important) modification. Due to the abundance of ss-exo enzymes in bacteria (PMID: 26442508), there is no such thing as ssDNA with unprotected ends there (the end polarity does not matter). Moreover, SSB binding, instead of protection, enhances ssDNA degradation by these processive enzymes (PMID: 16488881, PMID: 21572106). Therefore, stage III (SSB loading) cannot be correct. First, how come that SSB is loaded on the 3’-ending tail, but not on the 5’-ending tail? Second, both ss-tails will be SSB-complexed and immediately degraded by ss-exos, because the ends are still accessible. Actually, the authors got it (partially) right in Fig. 6A — it is the looping of the 3’-ending strand (via controlling the end) by AddA that prevents it from being degraded by ss-exos (the incorrect aspect is preservation of the unprotected strand). Therefore, I suggest this important AddA-looping to be introduced at stage II (end processing) and kept until stage V (RecA recruitment). I am not sure that, in *E. coli*, RecBCD release from DNA by RecA polymerization was ever demonstrated — I encourage the authors to contact Steve Kowalczykowski with this question. Thus, please remove all these unprotected ss-tails from stages III on (up to stage IX). Bacterial DNA metabolism simply does not tolerate them. Multiple ss-exo mutants are sick — the reason, although clear, is still unexplored experimentally.

We agree with the reviewer that there are many nucleases in the cell. However, we note that the publication supporting a role of SSB to facilitate ssDNA degradation reports an enzymatic reaction *in vitro*, therefore how this relates to SSB *in vivo* is unclear.

To further investigate this model of redundant nucleases, we deleted the major nuclease *Bacillus* nuclease RecJ in an AddAB nuclease-deficient background. We observed that viability remained unaffected upon introduction of nicks in the triple nuclease-deficient mutant (Fig. S12D). In live cells, and considering the evidence presented in this manuscript, we argue it is conceivable that during recombinational repair at a single-strand discontinuity, nuclease activity may not be essential (lines 275-283).

As suggested, in the proposed model (Fig. 8) we have included looping via AddA as an intermediate step prior to SSB loading on ssDNA (note that DNA is represented linearly to improve readability) and free DNA ends have been removed. We note that the Kowalczykowski lab has reported: “To date, the co-ordinated loading of RecA by the AddAB enzyme has not been demonstrated for the *B. subtilis* system (F. Chédin and S. C. Kowalczykowski, unpublished observations)” (PMID: 11929535).

(4) Page 19, line 459 — the lack of requirements for Holliday junction migration/resolution for restart of collapsed forks sounds paradoxical, — but in this work this is likely due to the discrepancy between what is measured by the two readouts in question. The colony formation after overnight plating with the inducer (Fig. 3A) tells us that HJ resolution is important over multiple generations and breaks, while the replication restart after one or two double-strand end repair events (Fig. 3B) shows that HJ resolution is not THAT important. In fact, the fork restart per se probably does not even require HJ resolution — as reflected in the classic models of the process (for example, Fig. 23 of PMID: 10585965). However, long-term functionality of the chromosome does — and this would be likely seen if the time of “mild” Cas9 induction was increased from one hour to several hours. This should be discussed in the following section.

In the revised manuscript, we have proposed that RecG helicase/translocase could stimulate branch migration of the Holliday junction under these conditions, rather than RuvAB (lines 480-484).

Minor points

Page 4, line 85 (also page 9, line 203, 206, 209 and so on, throughout the paper) — change “homologous recombination” to “recombinational repair”.

Changed throughout.

Pages 7-8, lines 165-173 — (this is FYI, not a request for more experiments) to actually distinguish between the fork stalling and fork restart, the same measurements for the helicase loaders should be done in the *priA* mutant, where the stalling could still happen, while reloading is blocked.

This is a great idea, although technically challenging in *Bacillus* because *PriA* is essential.

Page 9, lines 204-205 — this statement has to be modified to tone it down: cytology of the nucleoid cannot indicate specific DNA events — only that something dramatically bad has happened with the chromosome. Physical DNA studies could — in this case, PFGE could detect site-specific and one-sided double-strand breaks at the nicks.

This statement has been changed (lines 211-212): “Genetic and cytological data suggest that when a replisome encounters a nick, it generates a substrate with ssDNA that is sensed by RecA (Fig. 1C-E)”.

Page 9, line 207 — there is nothing wrong with your ref#39 (published in 2000), other than that it is just a synopsis for the general public of the 1996 monograph “Recombinational Repair of DNA Damage” by Kuzminov (ISBN: 0-412-10671-X), or its 1999 bacteria-only version

(PMID: 10585965). It is always better to cite the originals, — especially in this case, because the very phenomenon of replication fork instability (various mechanisms of disintegration) was recognized by Kuzminov (PMID: 7661854, PMID: 11459990), and the term “fork collapse” was also his contribution to the field (PMID: 8026461, PMID: 7565099).

Primary references have now been included.

Page 9, line 218 and Fig. S8D — The *ruvAB* mutants have a modest effect (reduced titer), similar to *recU*, so I would not dismiss them. You could tell it by their thinner-looking colonies.

We agree that *RuvAB* deficient strains display a mild growth defect upon nick induction (Fig. S8D, lines 228-230), however the aim of this study was to determine and characterise the most critical proteins for recombinational repair at a nick in *Bacillus*.

Page 10, lines 225-226, “the hypothesis that homologous recombination is necessary to repair the DSB” — this WAS the hypothesis in the 1960s and through 1970s, but was already an accepted concept in the 1980s (remember the famous model of DSB repair PMID: 6380756?). Say instead: “These observations are consistent with recombinational repair of disintegrated replication forks” (PMID: 10585965).

Restated throughout.

Page 10, lines 229-231 — References #50 and 51 are not relevant here. In *E. coli*, *RecFOR* proteins are dispensable for DSB repair, because *RecA* loading onto ssDNA with concomitant *SSB* displacement is done by the *RecBCD* enzyme itself (PMID: 10731409, PMID: 16483938, PMID: 38261972). Better cite papers showing that *RecFOR* functions are required for DSB repair in *Bacillus* (for example, PMID: 15186413).

We have added the recommended reference. The other references support the phrase (lines 245-247): “The single-stranded DNA (ssDNA) is recognised and bound by the single-stranded binding protein (*SSB*), which must be removed by the *RecFOR* complex to allow *RecA* filament formation (Fig. 3C(IV-V))”.

Page 10, line 233 — ref#53 is irrelevant to the statement it is supposed to support.

The reference was removed.

Page 10, line 237 — what is seDSB?

Wording was changed to single-end DSB (seDSB) and defined (lines 50-51).

Page 14, line 338 — change “well” to “(barely)” (in parenthesis). Compared with WT, the colonies are barely growing.

This section has been removed from the manuscript.

Page 15, line 364 — delete “this type of”. Is there any other type of collapsed forks? There are several types of fork disintegration events (for example, Fig. 6A of PMID: 24806348); fork collapse is the only one that happens at nicks.

Deleted as suggested.

Page 16, lines 393-396 — to see the effect of ss-exo deficiencies, one needs to inactivate several major ones in a single mutant, — inactivation of any one of them has either no phenotype or a weak one (at least in *E. coli*). Therefore, as explained above in major points, it is safe to assume that in bacteria, any unprotected ss-tail is immediately degraded, resulting in blunt ends, no matter which template strand is nicked.

Following this logic, we deleted the major exonuclease RecJ in a strain lacking AddAB nuclease activity. The results show that all three nuclease activities can be inactivated (Fig. S12D). While we cannot exclude the possibility that other nucleases act redundantly during recombinational repair in *B. subtilis*, our current model does not require nucleases. We have attempted to capture this nuance in the text (lines 280-283): “Based on these observations, we hypothesise that either another nuclease(s) promotes resection of the seDSB to generate a 3'-tail, or that AddAB helicase activity is necessary and sufficient to separate the two strands of the seDSB and facilitate SSB loading onto ssDNA with 3'-end”.

Page 17, lines 404-406 — I would modify the key role as to generate an end-protected ss-loop on the 3-ended strand (again, see above).

DNA looping via AddA is now shown in Fig. 8 and wording has been changed.

Page 17, line 407 — not only [SSB] is high near the fork, but the situation is likely the other way around: SSB forms a platform, on which the fork (or other processes, like AddAB-catalyzed unwinding) can function(see below).

Page 17, line 415 — yes, SSB is a protein hub, and this means it must form a platform for other proteins, as Lohman has long suggested. But since the bacterial nucleoid is a phase, this also means that the SSB platforms float on the surface of this phase and, besides handling ssDNA, serve as the nucleation points for assembly of secondary platforms, like the RecA one (PMID: 38358278). Fittingly, your SSB $\Delta 6$ mutation that should disrupt the formation of such a platform, has terrible effects on the subsequent recombinational repair (although it does not affect ssDNA binding).

We have modified this section accordingly and included relevant references.

Page 18, lines 430-433 — the RecF-DnaN interactions, if confirmed in *Bacillus*, will be relevant only for the daughter-strand gap repair. Here the authors are trying to apply these interactions to the double-strand end repair, but there is no replisome around — it was out of the picture the moment replication fork has collapsed. Therefore, I suggest the authors just point out that at the moment it is not clear what invites RecF to the AddAB-held ssDNA loop. I would bet on direct interactions between AddAB and RecF.

The possible role of DnaN recruiting RecF has been removed from the revised manuscript for clarity. However, we note that in *B. subtilis* it has been shown that approximately 200 DnaN clamps accumulate behind the replisome (PMID 21419346). Thus, it is reasonable to suggest that these DnaN proteins could play a role increasing the local concentration of RecF at sites near replication fork collapse.

Page 19, lines 455-457 — please clean up this word salad. Collapsed forks are not “resolved” (you do not have to repeat other people’s mis-statements), they need to be repaired and restarted (in prokaryotes) or just repaired (= the fork framework reassembled) without the restart (in eukaryotes, where no restart machinery exists). Further, replisome cannot collapse, only the fork can — and then the replisome had to lose contact with the resulting DNA pieces (and is likely disassembled).

We thank the reviewer for helping us with correct nomenclature and have amended the text throughout the manuscript (this specific section of concern was removed).

Page 32, line 753 — the helicase is shown as a circle, rather than hexagon. Also, in the panel G itself, replace “Helicase collapses” (what is this?) with “Helicase runs off”.

Wording has been changed and helicase is now consistently represented as a circle throughout figures.

All chromosome profiles with active nicking — how would the authors explain that the profile minima coincide with the position of the nick? Is it a real degradation of the template DNA around the nick? Or is it incomplete isolation of DNA for MFA, due to the region being complexed with recombinational repair proteins/enzymes?

This is an insightful question and the answer is we do not know. It may be a technical issue or it may reflect end processing. We are currently developing a method to probe strand specific DNA resection, and we hope to test these exciting hypotheses using this system.

Fig. 3B — for comparison purposes, I would repeat there the panels 2H (WT) and 2K (priA). Besides, there is space there for two more panels.

Amended as suggested.

Fig. 5B scheme is confusing, rather than clarifying. For example, do these events happen after fork collapse and repair? Or independent of it, at an intact replication fork? Fig. S11 does not help either (besides, the two structures should be the same, right?).

As suggested this figure has been removed from the revised manuscript.

Fig. 8C — “addO” is likely “recO”

This has been corrected.

Reviewer #3 (Remarks to the Author):

This is a very interesting and important set of experiments that define some of the events that occur when a bacterial replication fork hits a template discontinuity. The work is innovative, provides new information, and greatly solidifies speculation based on other, less direct observations. In general, the work provides a broader and much clearer view of a very important bacterial process. The proteins defined as essential for double strand break repair are not surprising but some of the proteins that are NOT required may surprise. The efforts to define RecG participation break some significant new ground and the observation of a RecO-SSB interaction requirement is nicely done. The absence of a requirement for the AddAB nuclease activities is also an important contribution. The system the authors have developed is quite complex and more effort is needed to lay out all the moving parts clearly. Most of my comments are to that end.

Fig 1C When the nickase is induced in normal cells, viability declines by some 5 orders of magnitude. But a few cells survive. In those cells, has the nickase been inactivated or the target site mutated?

The reviewer has a good eye. These suppressor mutants typically disrupt the promoter sequence of the sgRNA or the protospacer sequence itself. A statement has been added in the Methods to make the reader aware of technical challenges that can arise from working with nucleases in live cells (lines 604-606): "Sanger sequencing identified that the strong nCas9/sgRNA nicking system is prone to accumulation of suppressor mutations either disrupting protospacer or its promoter sequence (Fig. S1F)".

Fig 2A. Some distance indication is needed. How far from the nick site are these various positions? There are many references to upstream and downstream of the nick. How far upstream and downstream in the exp of Fig 2?

Relative distances have been clarified (Figs. 2A, S4).

Fig S6B. Is xylose present here?

Xylose is absent in this panel. Figure legend and description of the *priA* complementation system have been clarified in relevant sections (Results and Methods).

In coli, RecBCD is able to load RecA on its own and the requirement here for RecFOR may surprise some readers. If AddAB does not load RecA on its own, a discussion of this difference would be useful. Also, a mention of RecU function when it first appears would be useful.

We note that the Kowalczykowski lab has reported: "To date, the co-ordinated loading of RecA by the AddAB enzyme has not been demonstrated for the *B. subtilis* system (F. Chédin and S. C. Kowalczykowski, unpublished observations)" (PMID 11929535).

RecU function has been described (lines 251-252): "RecU is a structure specific nuclease involved in cleaving the Holliday junction generated by RecA (Fig. 3C(VII))".

RESPONSE TO REVIEWERS: NCOMMS-25-28666A (R2)

Reviewer #1 (Remarks to the Author):

I would like to thank the authors for carefully addressing all the comments and providing additional data that strengthen the manuscript. Without listing all the changes, I was particularly impressed by the experiment involving the RexAB protein of *S. aureus*. The observation that the SSB recruitment position is altered, aligning with the position of the *S. aureus* Chi sites, strongly supports the authors' conclusions. Furthermore, the rewriting has greatly clarified the text.

I congratulate the authors on an excellent paper reporting very exciting results!

My only minor suggestion is to use the term “the replicative helicase” in the abstract (line 25) instead of “helicase” to avoid confusion.

Meriem El Karoui

We modified wording as suggested.

Reviewer #2 (Remarks to the Author):

Good job, authors! I really enjoyed this paper.

Minor points

Page 12, lines 281-283: “... or that AddAB helicase activity is necessary and sufficient to separate the two strands of the seDSB and facilitate SSB loading onto ssDNA at a 3'-end.” — This is unlikely, as the 5'-ending strand still needs to be removed — otherwise it will poison the subsequent RecA-mediated homology search by providing the proximal “homology”. It will be preferentially removed by (unspecified) ssDNA exonucleases, as long as the 3'-ending strand is secured in a loop by AddAB.

As suggested, we modified this statement (lines 284-286): “Based on these observations, it appears that under these conditions AddAB helicase activity and an unspecified nuclease promote end-processing of the seDSB to generate a 3'-tail”.

Page 12, lines 286/287: “The requirement of the RecFOR system for recombinational repair (Fig. 2) implied that SSB is bound to the strand terminating with a 3'-end.” — Why the 3'-end? I am not sure where is this specific polarity coming from. Hypothetically, it could be through the AddAB-RecFOR physical interactions, but there is no evidence about it yet.

We changed wording to aid clarity.

Page 20, lines 490-494 — these vague explanations of the Δ recU mutant defects in chromosome segregation are completely unnecessary. As I pointed out in the previous round of critique, restart of collapsed replication forks after reassembly by recombinational repair was always proposed to be independent of the Holliday junction resolution behind the fork (for example, Fig. 23 of PMID: 10585965), but Holiday junction resolution is required to finish the

repair — otherwise the two sister chromosomes will remain linked together and unsegregatable! And the XerCD monomerization system will not help in this case.

In fact, the authors should argue in Discussion that this result supports the idea that fork restart is independent of HJ resolution, but HJ resolution is required for completion of repair via disconnecting the sister chromatids — otherwise chromosome partitioning problems.

This paragraph was rewritten as suggested (lines 494-501): “While *recU* was identified as important [...] these results support the model that recombinational repair is mechanistically distinct from resolution of conjoined chromosomes”.

Miscellaneous

Page 16, line 375 — change “genome” to “template DNA”. “Genome” refers to information content of DNA, rather than to DNA structure.

Done.

Page 16, line 381 “intolerant” — is “tolerant” meant there?

Corrected as suggested.

Pag 18, line 449 “replication repair” — was “recombinational repair” meant here?

Andrei Kuzminov

Corrected as suggested.

Reviewer #3 (Remarks to the Author):

The response to referees is comprehensive and a strong study has been rendered stronger as a result. However, one error has been added in the revision that should be removed. On line 246, the authors have now mentioned a RecFOR complex. There is no such thing and its mention can be misleading. As no RecFOR complex has ever been described (there are RecOR or RecFR complexes but no RecFOR), the word “complex” here needs to be changed to “system”.

We changed wording as recommended.